**Investigation**

# On the patterns of genetic intra-tumor heterogeneity before and after treatment

Alexander Stein ⓘ ,* Benjamin Werner ⓘ *

Evolutionary Dynamics Group, Centre for Cancer Evolution, Barts Cancer Institute, Queen Mary University of London, Charterhouse Square, London EC1M 6BQ, United Kingdom

*Corresponding author: Evolutionary Dynamics Group, Centre for Cancer Evolution, Barts Cancer Institute, Queen Mary University of London, Charterhouse Square, London EC1M 6BQ, United Kingdom. Email: alexandersteinresearch@gmail.com; *Corresponding author: Evolutionary Dynamics Group, Centre for Cancer Evolution, Barts Cancer Institute, Queen Mary University of London, Charterhouse Square, London EC1M 6BQ, United Kingdom. Email: b.werner@qmul.ac.uk

Genetic intra-tumor heterogeneity is a universal property of all cancers. It emerges from the interplay of cell division, mutation accumulation, and selection with important implications for the evolution of treatment resistance. Theoretical and data-driven approaches extensively studied intra-tumor heterogeneity in ageing somatic tissues or cancers at detection. Yet, the expected patterns of intra-tumor heterogeneity during and after treatment are less well understood. Here, we use stochastic birth–death processes to investigate the expected patterns of intra-tumor heterogeneity across different treatment scenarios. We consider homogeneous treatment response with shrinking, growing, and stable disease, and follow-up investigating heterogeneous treatment response with sensitive and resistant cell types. We derive analytic expressions for the site frequency spectrum, the total mutational burden and the single-cell mutational burden distribution that we validate with computer simulations. We find that the site frequency spectrum after homogeneous treatment response retains its characteristic power-law tail, while emergent resistant clones cause peaks corresponding to their sizes. The frequency of the largest resistant clone is subdominant and independent of the population size at detection, whereas the relative total number of resistant cells increases with detection size. Furthermore, the growth dynamics under treatment determine whether the total mutational burden is dominated by preexisting or newly acquired mutations, suggesting different possible treatment strategies.

Keywords: cancer evolution; intra-tumor heterogeneity; treatment resistance; birth–death processes

## Introduction

Every cell in a human body accumulates thousands of unique mutations with age (Abascal *et al.* 2021; Mitchell *et al.* 2022). From this genetic heterogeneity, premalignant cells may arise and eventually evolve into an expanding cancerous cell population. As the disease progresses, cancer cells continue to accumulate mutations, subject to genetic drift and selection, leading to a complex pattern of genetic intra-tumor heterogeneity (gITH) (Greaves and Maley 2012; Turajlic *et al.* 2019). Mathematical and computational modeling applied to sequencing data of normal and cancerous tissues has led to a better understanding of the emerging patterns of clonal diversity (Simons 2016; Williams *et al.* 2018; Caravagna *et al.* 2020; Watson *et al.* 2020; Moeller *et al.* 2024). It is clear that gITH informs on past evolutionary paths and at least partially determines future evolution, e.g. the risk of resistance or relapse in treated tumors (Andor *et al.* 2016 ; Fernandez-Mateos *et al.* 2024). While patterns of gITH in both normal and cancerous cell populations prior to treatment are well studied, less is known about the dynamics of gITH in response to treatment.

A common way to quantify gITH is based on the distribution of cellular abundances of all detectable mutations. This statistic is known as the site frequency spectrum (SFS), which is closely related to the variant allele frequency (VAF) spectrum that can be obtained from single-cell or bulk sequencing data (Williams *et al.* 2016; Moeller *et al.* 2024). The SFS has been mathematically well characterized for exponentially growing populations using coalescent theory (Griffiths and Tavaré 1998; Ohtsuki and Innan 2017) and birth–death processes (Durrett 2013; Bozic *et al.* 2016; Ohtsuki and Innan 2017; Gunnarsson *et al.* 2021, 2025). Importantly, the tail of the SFS in an exponentially growing population is known to follow a power law with exponent 2, while the SFS of a constant population at equilibrium is known to follow a power law with exponent 1 (Griffiths and Tavaré 1998; Durrett 2008; Gunnarsson *et al.* 2021). These scaling laws are observed in bulk sequencing data of cancers (Williams *et al.* 2016) and healthy somatic tissues in homeostasis (Moeller *et al.* 2024).

Starting with the pioneering work of Luria and Delbrück (1943) that combined mathematical modeling and experiments to decipher if resistance of bacteria to bacteria-phage was emerging or preexisting, a considerable body of theoretical and experimental literature arose estimating the number of resistant cells in an exponentially growing population (Zheng 1999; Rosche and Foster 2000). The Luria–Delbrück distribution, i.e. the size distribution of resistant cells, was originally studied in a semi-deterministic model with random mutations and deterministic growth (Luria and Delbrück 1943). Shortly after, the model was extended to stochastically growing mutants that follow a pure-birth process (Lea and Coulson 1949). More recently, a number of approximate and

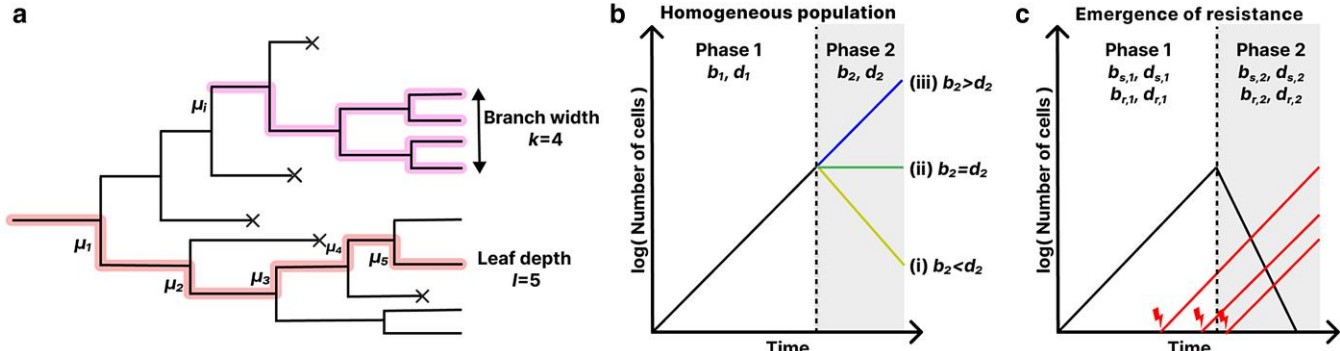

**Fig. 1.** Model illustration. a) Phylogeny with a highlighted branch (purple) and highlighted path from leaf to root (red). b) Growth of homogeneous population: In phase 1, the population grows exponentially. In phase 2, the population (i) decreases, (ii) remains constant, or (iii) continues increasing. c) Growth with the emergence of resistance: In phase 1, the cancer grows exponentially and resistance mutations lead to a treatment-resistant subpopulation. In phase 2, the sensitive population decreases while the resistant population continues increasing.

exact solutions were obtained for the fully stochastic birth–death process conditioned on fixed population size or fixed time (Antal and Krapivsky 2011; Kessler and Levine 2013, 2015; Cheek and Antal 2018, 2020). The SFS and Luria–Delbrück distribution are two different quantities per se, which are however mathematically related. Given the equivalence of mutation rates for specific sites and mutation rates to obtain resistance, Cheek and Antal (2018, 2020) showed explicitly that the expected SFS can be obtained from a generalization of the Luria–Delbrück distribution. Alternatively, if per-division mutation rates are sufficiently low, the clone size distribution (CSD) of the resistant subpopulation is well approximated by the expression of the SFS.

When summing over all clonal and subclonal mutations in the population, we arrive at the total mutational burden (tMB). The exact tMB is difficult to estimate from sequencing data because most mutations are expected to be at very low frequencies that are below current sequencing resolution. However, even with limited resolution, the tMB can be used to infer evolutionary parameters (Lee and Bozic 2022). The tMB is one of the simplest theoretical measures of gITH. It is directly linked to the probability of resistance and can be derived from the SFS (Bozic *et al.* 2016 ; Gunnarsson *et al.* 2021 , 2025).

Instead of counting the cellular abundance of mutations, one can count how many mutations are present in single cells. Following Moeller *et al.* (2024), we call the resulting distribution the single-cell mutational burden (scMB) distribution. Recently, we have shown that the scMB distribution can be obtained from single-cell information, and it contains information on somatic evolutionary processes complementary to the SFS (Moeller *et al.* 2024). Assuming that mutations emerge during cell divisions and neglecting background mutations, the scMB distribution is directly coupled to the cell divisional distribution (Morison *et al.* 2023), which is known to converge to a normal distribution for long times (Bühler 1971; Samuels 1971) and is subject to an inspection bias (Cheek and Johnston 2023).

The properties and evolution of the SFS, tMB, and the scMB in the absence and presence of treatment are of practical interest because they may inform on the underlying process of emerging resistance and at the same time are obtainable in clinical studies. With the exception of Bonnet and Leman (2024) and Leder *et al.* (2024) that only considered mutations emerging after detection, a thorough theoretical assessment of the dynamics of the SFS, the tMB, and the scMB distribution during treatment is still lacking.

In this study, we use birth–death processes with the random accumulation of mutations to investigate gITH in cancers before

and after treatment. Building upon the results of the SFS and tMB at detection, we derive novel analytic expressions of the SFS and the tMB under different treatment scenarios. We refine the earlier results on the scMB distribution at detection and homeostasis and apply them to the treatment setting. Treatment is implemented as an abrupt change of the growth. For each measure of gITH, we (i) start with the mathematical construction, (ii) recapitulate results of exponential growth, and (iii) investigate the dynamics under homogeneous treatment response. We then make the first steps in assessing heterogenous treatment responses, where the emergence of resistance leads to a multiclonal disease subtly changing properties of the SFS and scMB.

## Theoretical framework

We model the cancer cell population using a birth–death process with constant birth and death rates. Starting from a single ancestor cell, cells either divide or die, giving rise to a phylogeny (Fig. 1a). Initially, the birth rate is larger than the death rate ($b_1 > d_1$) leading to a growing population (phase 1). Once the cancer is detected at time $t_d$ with size $N_d$, treatment is administered leading to new birth and death rates $b_2$ and $d_2$ (phase 2). Treatment is applied for time $t_f$ after which we have the final population size $N_f$. Assuming a homogeneous treatment response, i.e. all cells have the same birth and death rates during treatment, there are three qualitatively different scenarios for the growth: The cancer decreases ($b_2 < d_2$ and $N_d > N_f$), remains approximately constant ($b_2 = d_2$ and $N_d \approx N_f$) or continues growing ($b_2 > d_2$ and $N_d < N_f$) as illustrated in Fig. 1b.

We investigate the emergence of resistance by dividing the population into treatment-sensitive and treatment-resistant cell types. We start with a single sensitive progenitor cell. When a sensitive cell divides, each daughter cells acquires a resistance mutation at rate $\nu$. The resistance mutation will be inherited to the offspring giving rise to a clone with new birth and death rates (Fig. 1c). In total, we have 8 growth parameters: the birth and death rate of sensitive cells before treatment $b_{s,1}, d_{s,1}$ and during treatment $b_{s,2}, d_{s,2}$, and the same set of parameters again for resistant cells $b_{r,1}, d_{r,1}$ and $b_{r,2}, d_{r,2}$. Unless otherwise mentioned, we assume that resistant cells behave neutral in the absence of the drug ($b_{s,1} = b_{r,1}$ and $d_{s,1} = d_{r,1}$), sensitive cells decrease in the presence of treatment ($b_{s,2} < d_{s,2}$) while resistant cells continue to increase ($b_{r,2} > d_{r,2}$).

To model the accumulation of neutral mutations, we assume that in each division, each cell obtains $\mu$ mutations, where $\mu$ is a

random number drawn from a Poisson distribution with mean $m$. Since each division comes with 2 daughter cells, there are on average 2 m mutations generated per division. We assume infinite sites such that each mutation is treated unique and mutation reversion is neglected. In Supplementary Section P, we discuss the biological and theoretical rationale. We make connections to the cases in which mutations are accumulated on only one of two daughter cells during division or in which maximal one mutation is accumulated with a fixed probability rather than a Poisson process.

We denote the stochastic total number of cells by $Z_0(t)$, its expectation by $N(t) = E[Z_0(t)]$ and add a tilde when conditioned on survival $\tilde{N}(t) = E[Z_0(t)|Z_0(t) > 0]$. Next to the total population, subpopulations will follow stochastic growth too. We write $p(a \to n, t)$ for the probability that a (sub-)population has size $n$ after time $t$ when starting from size $a$. In its most general form, $p(a \to n, t)$ is a sum over $\min(a, n)$ terms that yield little analytic insight and is computationally expensive (Tavaré 2018; Davison *et al.* 2021). Therefore, we synthesized exact and approximate results to balance computational efficiency and accuracy (Supplementary Section A and Materials and Methods).

Some cancers, even if they have positive fitness, go extinct. If we observe a cancer, it means that it survived genetic drift, which is mathematically incorporated by conditioning on survival. We denote the probability for a population of size 1 to go extinct by time $t$ with $\alpha(t)$. Given birth rate $b$ and death rate $d$, the extinction probability for a population of size $a$ is then

$$p(a \to 0, t) = \alpha^a(t) \quad \text{with} \quad \begin{cases} \alpha(t) = \dfrac{de^{(b-d)t} - d}{be^{(b-d)t} - d} & \text{if } b \neq d, \\[2mm] \alpha(t) = \dfrac{bt}{1 - bt} & \text{if } b = d. \end{cases} \tag{1}$$

We can then write the average growth conditioned on survival as

$$\tilde{N}(t) = \frac{a}{1 - \alpha^a(t)} e^{(b-d)t}, \tag{2}$$

whereas the average growth including extinct trajectories is $N(t) = ae^{(b-d)t}$ (Kendall 1948 ; Durrett 2015).

Observations are made either at a fixed size or a fixed time. If we stop the stochastic process once the population reaches a fixed time, we have a variable population size. The random size is characterized by $p(a \to n, t)$. Vice versa, if we stop at a fixed size, time will be a random variable. Conditioned on survival, the random time $T_N$ to grow to size $N$ for the first time when starting from a single cell is known to approximately follow a Gumbel distribution (see Durrett 2015 and Supplementary Section B) with expectation

$$E[T_N] = \frac{1}{b - d} \left( \ln\left(\frac{b - d}{b} N\right) + \gamma \right), \tag{3}$$

where $\gamma \approx 0.5772$ is the Euler–Mascheroni constant. Both conditions have relevance in biological systems: In the clinic, the starting time of the tumor is unknown and we observe the disease once it reaches detectable size. In contrast, in laboratory set-ups, experiments start and end at predefined times. It is possible to switch the conditioning of predictions between fixed time and fixed size (Supplementary Section C). Using appropriate mapping between fixed time and fixed size conditions, we will see that expected values of the considered statistics are very close to each other.

We investigate gITH of the cell population in terms of three summary statistics. First, we look at the SFS. The SFS distributes mutations into classes $\mathcal{S}_k$ according to the number of cells $k$ they occur in. The sizes of the classes $S_k = |\mathcal{S}_k|$ build a distribution $\{S_k\}_{k=1}^{N}$ that we call the SFS. A site frequency corresponds to the abundance $k$ of a mutation that resides at a particular site. Second, we study the tMB that we define as the number of unique mutations in the entire cancer and denote by $B$. The tMB counts mutations that are either clonal or subclonal and is related to the SFS by $B = \sum_{k=1}^{N} S_k$. Third and last, we compute the scMB distribution. The scMB is the number of mutations in a single cell. The scMB distribution, $\{M_j\}_{j=1}^{\infty}$, is defined by the number of cells, $M_j$, with $j$ mutations. In this sense, the scMB distribution distributes cells into classes according to the amount of mutations they carry.

Summary statistics can be computed from a single realization, and generally differ between different realizations. From this perspective, the SFS $\{S_k\}_{k=1}^{N}$ consists of $N$ random variables, the tMB $B$ consists of 1 random variable, and the scMB distribution $\{M_j\}_{j=1}^{\infty}$ consists of infinitely many random variables. Mathematically speaking, the SFS and the scMB distribution are random distributions, i.e. distributions in which each value is a random variable. In the following, we are mainly interested in their expectations but also present some results of the variance between realizations. The latter corresponds to inter-tumor heterogeneity rather than gITH.

## Results

### The site frequency spectrum

The SFS can be derived from the VAF spectrum that is frequently obtained from bulk whole-genome sequencing (Williams *et al.* 2016). Here, we link the SFS to the phylogenetic tree emerging from the birth–death process. We build up on the results of Gunnarsson *et al.* (2021) on the expected SFS at detection to derive analytic expressions of the expected SFS after a homogeneous treatment response. Based on our analytic results, we suggest a null hypothesis for homogeneous treatment response that can be tested on sequencing data.

#### Construction of the site frequency spectrum

Inspecting the phylogenetic tree in Fig. 1a, we see that the number of mutations $S_k$ that occur in $k$ cells is obtained by summing over all mutations with branch width $k$. If we write the number of branches with width $k$ as $W_k$, label the branches with width $k$ by $j$ and the number of mutations on branch $j$ by $\mu_{k,j}$, we have

$$S_k = \sum_{j=1}^{W_k} \mu_{k,j}. \tag{4}$$

Since we assume infinite sites, the number of mutations $\mu_{k,j}$ are independent and by assumption follow a Poisson distribution, $\mu_{k,j} \sim \text{Poiss}(m)$. We call the collection $\{W_k\}_{k=1}^{N}$ the branch width distribution, which characterizes the tree topology that emerged from the stochastic growth of the underlying cell population. Since equation (4) is a random sum in which $\mu_{k,j}$ are independent and identical distributed, we can write the expectation as

$$\begin{aligned} E[S_k] &= E[W_k]E[\mu_{k,1}] \\ &= E[W_k]m \end{aligned} \tag{5}$$

and the variance as

$$\begin{aligned} Var[S_k] &= Var[\mu_{k,1}]E[W_k] + E^2[\mu_{k,1}]Var[W_k] \\ &= mE[W_k] + m^2 Var[W_k], \end{aligned} \tag{6}$$

which disentangle the effects of stochastic growth and mutation accumulation. As we shall see in further proceedings, it is useful to separate between mutations that are preexisting to time $t' = 0$ or newly emerging in a time interval $[0, t]$. We denote the number of newly emerging mutations found in $k$ cells by $S_k^{(new)}$, and the number of preexisting mutations by $S_k^{(pre)}$. The total number of mutations found in $k$ cells is then $S_k^{(tot)} = S_k^{(new)} + S_k^{(pre)}$. The following theorem characterizes the expected SFS of both contributions.

**Theorem 1** Suppose the expected total population size over the time interval $[0, t]$ is $N(t')$ with initial size $N_0$, and the probability to grow from size $x$ to $y$ in time $t$ is $p(x \rightarrow y, t)$. Then, the expected SFS of newly emerging mutations conditioned on survival and fixed time $t$ is

$$E[S_k^{(new)}] = \frac{1}{1 - p(N_0 \rightarrow 0, t)} \int_0^t 2mbN(t')p(1 \rightarrow k, t - t')dt'. \quad (7)$$

Suppose the expected initial SFS at time $t' = 0$ is $E[S_{k'}^{(init)}]$, then the expected SFS of preexisting mutations conditioned on survival and fixed time $t$ is

$$E[S_k^{(pre)}] = \frac{1}{1 - p(N_0 \rightarrow 0, t)} \sum_{k'=1}^{N_0} E[S_{k'}^{(init)}] p(k' \rightarrow k, t). \quad (8)$$

*Proof.* Supplementary Section D.

For the proof of equation (7), we follow Gunnarsson *et al.* (2021) who derived the result for $N_0 = 1$. In both expressions, the first term is due to condition on survival of the entire population and can be dropped should one be interested in the expectation including extinction. Intuitively, equation (7) counts the number of mutations generated at time $t'$, conditions them to grow to size $k$ and then sums over all mutations generated in the considered time interval. Equation (8) has similarities to a sampling formula. There are $E[S_{k'}^{(init)}]$ mutations of size $k'$ that are "sampled" to size $k$ with probability $p(k' \rightarrow k, t)$. We do not strictly talk about sampling since we allow $k > k'$.

### The site frequency spectrum at detection
The expected SFS at detection is computed by applying Theorem 1 over the time interval $[0, t_d]$ starting from a single cell with no mutations (Supplementary Section E). Following Ohtsuki and Innan (2017) and Gunnarsson *et al.* (2021), the resulting integral can be simplified to

$$E[S_k^{(det)}] = 2mN_d \int_0^{1-1/N_d} \left(1 - \frac{d_1}{b_1}y\right)^{-1} (1 - y)y^{k-1} \, dy$$
$$\rightarrow 2mN_d \sum_{l=0}^{\infty} \frac{\left(\frac{d_1}{b_1}\right)^l}{(k+l)(k+l+1)} \quad \text{for } N_d \rightarrow \infty \quad (9)$$
$$\rightarrow 2mN_d \frac{b_1}{b_1 - d_1} \frac{1}{k(k+1)} \quad \text{for } k \rightarrow \infty.$$

Here, $N_d = \tilde{N}(t_d)$ is the expected population size at time $t_d$ conditioned on survival. Although not the focus of this study, it is straightforward to include mutations accumulated before expansion that will be clonal in the entire cancer (Supplementary Section E).

The prediction in equation (9) perfectly aligns with the average SFS obtained through computer simulations (Fig. 2a). For large $k$, we have $E[S_k^{(det)}] \sim k^{-2}$, which is a well-known characteristic of the SFS in exponentially growing populations observable in bulk sequencing data of sufficient coverage (Williams *et al.* 2016, 2018; Caravagna *et al.* 2020). Noticeably, single realizations of the stochastic process deviate significantly from the expected scaling for large site frequencies (Fig. 2a, dots). These are mutations occurring in the first few generations of growth and have a high degree of stochasticity.

Knowing the probability density $f_T(t|N_d')$ for the time $t$ to reach fixed size $N_d'$ for the first time (Supplementary Section B), we can change the conditioning from fixed time to fixed size (Supplementary Section C). To avoid convergence issues, we consider the normalized SFS that we denote with a tilde $E[\tilde{S}_k|N_d']$. We change the conditioning by computing

$$E[\tilde{S}_k|N_d'] = \int_0^{\infty} E[\tilde{S}_k|t] f_T(t|N_d') \, dt, \quad (10)$$

Using the large population size solution $N_d \rightarrow \infty$ in equation (9), we show that the fixed-size expectation coincides with the fixed-time solution, $E[\tilde{S}_k|N_d'] = E[\tilde{S}_k|T = t]$ (Supplementary Section E). This is in agreement with Gunnarsson *et al.* (2021) who has shown that the fixed-size expectation can be approximated by the fixed-time expectation for large $N_d$.

The expected SFS after exponential growth is well studied. However, with exception of Cheek and Antal (2020), the variance of the SFS under exponential growth is not considered. In the pure-birth process, and within the fixed-size limit, we observe in simulations that $W_k$ are approximately Poisson distributed such that $S_k$ is described by a compound Poisson process (Fig. 2b). This suggests a heuristic expression for the variance

$$Var[S_k^{(det)}] = (1 + m)E[S_k^{(det)}] \quad (11)$$

that shows good agreement with simulations of the pure-birth process on fixed size (Fig. 2c). There are deviations for very small site frequencies $k = 1, 2, 3 \ldots$. To better understand this, we point out that when conditioning on growth to size $N_d$, there are exactly $N_d$ branches, which are simultaneously leaves such that $Var[W_1^{(det)}] = 0$ and thus $Var[S_1^{(det)}] = mE[W_1^{(det)}]$. Interestingly, equation (11) suggests that the variance in the SFS carries information on the mutation rate $m$. This can be leveraged to infer the mutation rate $m$ from sequencing data, possibly at different stages of tumor growth.

### The site frequency spectrum after treatment
We compute the SFS after a homogeneous treatment response, and distinguish between decreasing, constant and continued increasing cell populations (Fig. 1b). To apply Theorem 1, we assume that the cancer has initially fixed size $N_d'$ and expected SFS at detection, $E[S_k^{(init)}] = E[S_k^{(det)}]$. For analytical purposes, we approximate the fixed-size expected SFS with the fixed-time expected SFS in equation (9). Together with the expected growth $N(t') = N_d' e^{(b_2 - d_2)t'}$, the treatment time interval $[0, t_f]$ and probabilities $p(a \rightarrow n, t)$ (Supplementary Section A), we can readily compute the expected SFS of newly emerging and preexisting mutations. We validate the general analytic predictions with simulations showing perfect alignment (Fig. 2d–f), and proceed with highlighting some approximations and observations.

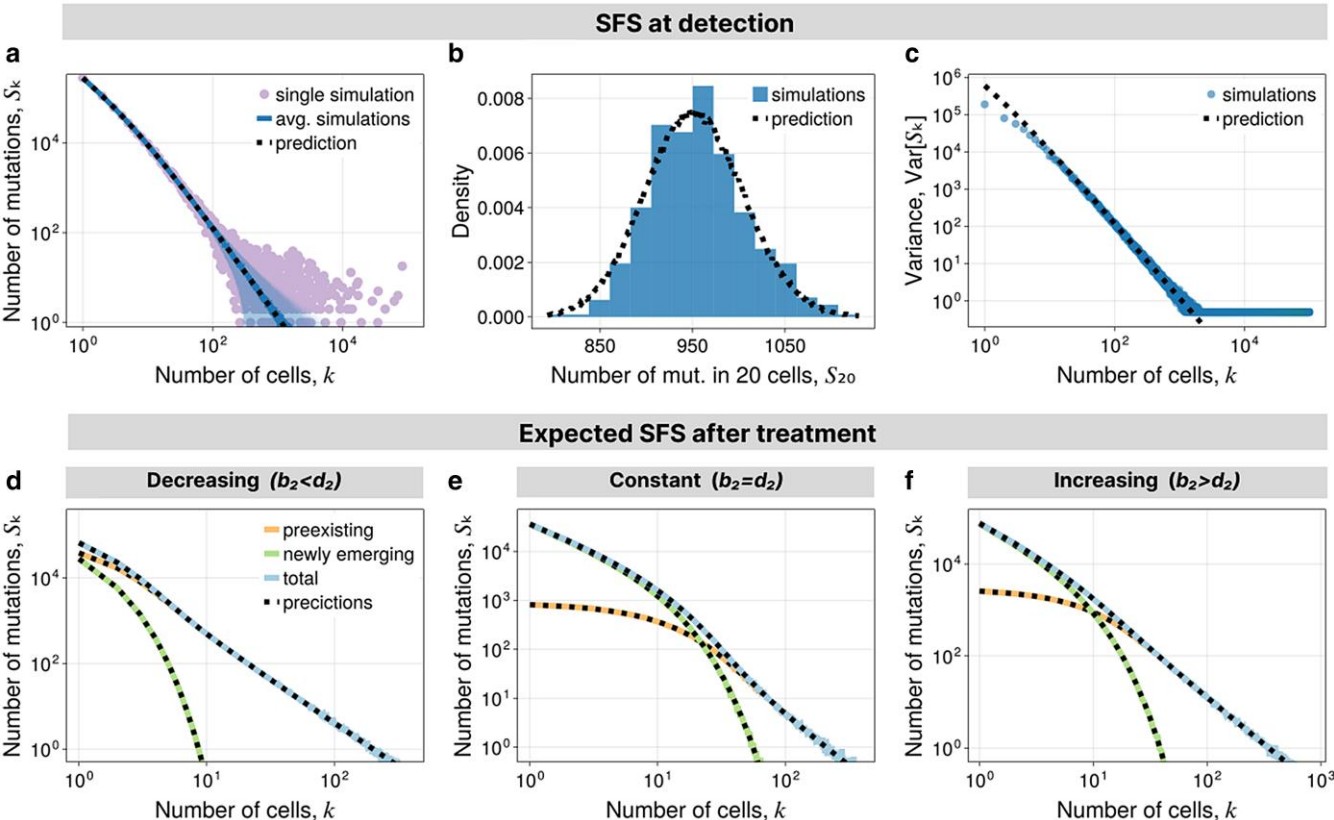

**Fig. 2.** SFS in homogeneous populations. a–c) SFS at the time of detection. a) Single simulation, average over many simulations with 1-sigma interval, prediction from equation (9). b) Distribution of $S_{20}$ from simulations and prediction using a compound Poisson distribution. c) Variance in $S_k$ from simulations and prediction from equation (11). d–f) SFS after treatment. Average over many simulations and prediction from Theorem 1 with explicit formulas in Supplementary Section F. d) For decreasing population. e) For constant population. f) For increasing population. Parameters are in Supplementary Table S1.

First, we observe that newly emerging mutations are restricted to small site frequencies, and thus the tail is determined by preexisting mutations. In shrinking populations, newly emerging mutations are prone to extinction and only survive short times and thus have almost negligible impact on the SFS. In contrast, for constant or increasing tumor populations, newly emerging mutations dominate the SFS at low frequencies, whereas preexisting mutations continue to dominate the SFS at high frequencies (Fig. 2d–f).

For decreasing populations, we obtain a simple formula in the case of the pure-death process. To show this, we adapt the sampling process from Theorem 3 from Durrett (2015) (Supplementary Section G). Taking $b_2 = 0$ and sufficiently large $k$, we find

$$E[S_k^{(\text{tot})}] = E[S_k^{(\text{pre})}] \approx 2mN_f \frac{b_1}{b_1 - d_1} \frac{1}{k^2 + k}, \tag{12}$$

where $N_f = \tilde{N}(t_f)$ is the expected population size after treatment for time $t_f$ conditioned on survival. There are no newly emerging mutations during treatment since $b_2 = 0$, and the $k^{-2}$ tail of neutral mutations from the SFS at detection is maintained.

As $b_2 \to d_2$, the importance of genetic drift increases, and if $b_2 = d_2$, the SFS of newly emerging mutations takes the form

$$E[S_k^{(\text{new})}] = 2mN_f \left( \frac{b_2 t_f}{b_2 t_f + 1} \right)^k \frac{1}{k}. \tag{13}$$

As $t_f \to \infty$, all preexisting mutations will be either extinct or fixed in the population such that only newly emerging mutations

contribute to gITH. From equation (13), we can see that the SFS scales with $k^{-1}$ in agreement with previously reported results for the SFS in constant populations at equilibrium (Griffiths and Tavaré 1998; Durrett 2008; Gunnarsson et al. 2021).

## Testing homogeneity

Theorem 1 provides us predictions for the SFS at detection and after treatment assuming that all cancer cells have the same growth parameters. In the spirit of the neutral theory of molecular evolution (Kimura 1968, 1991), we can take this prediction as null hypothesis of homogeneity that can be compared to sequencing data as done by Williams et al. (2016). If the population is homogeneous, we expect a good fit between the data and the prediction. If the population is heterogeneous, we expect to see deviations between data and prediction.

Sequencing data are noisy and mutations that occur in <1% of the cell population cannot reliably be observed (Stead et al. 2013). This restricts our test on the observable mutations occurring in at least 1% of the population ($\frac{k}{N_f} > 0.01$). However, considering large site frequencies dramatically simplifies the homogeneity test in the treatment scenario. From simulated data, we observe that the SFS in mutations above 1% of the population follow the power-law $S_k \sim k^{-2}$ known from exponentially growing tumors (Supplementary Figs. S1 and S2). We support this observations by arguing that the observable SFS for biologically feasible parameters consists of large site frequencies that behave nearly deterministic (Supplementary Section G). Remarkably, this holds true for all three considered cases of homogeneous growth, although

deviations may occur for small populations that are kept constant for long times (Moeller *et al.* 2024). We conclude that testing homogeneity can be achieved by evaluating whether the SFS follows the expected scaling $E[S_k^{(tot)}] \sim k^{-2}$ as demonstrated in prior analyses of the SFS at detection (Williams *et al.* 2016, 2018).

In sequencing data, not the SFS of the population but the VAF spectrum of a sample is measured. Let us consider a sample of $n$ cells. The VAF spectrum of the sample is obtained by counting the number of mutations $V_k$ with abundance $k$. Abundances $k$ are normalized to frequencies $f = \frac{k}{n}$ leading to $\tilde{V}_f$, which is typically presented in data-driven studies. Assuming that there are no copy-number changes, each cell has 2 alleles corresponding to one site. Mutations occur usually only in one of the two alleles such that a mutation present in all cells corresponds to an allele frequency $f = \frac{1}{2}$. Consequently, the number of mutations found in $k$ cells, i.e. the SFS, can be retrieved by computing $S_k = V_{2k}$. Different bioinformatic tools can be used to estimate copy-number changes, which also correct for sample purity (Van Loo *et al.* 2010; Nik-Zainal *et al.* 2012). One can then either focus on regions in the genome that do have no copy number alterations or try to account for the copy-number evolution (Tanner *et al.* 2021). In the case of aneuploidies, the raw VAF spectrum demonstrates additional peaks, which can be identified and corrected (Caravagna *et al.* 2020). Notably, testing homogeneity is restricted due to sampling biases and model assumptions that are delineated in the Discussion.

## The total mutational burden

The tMB is difficult to estimate from sequencing data. The majority of mutations occur at low frequencies and are not detectable with bulk sequencing methods. However, an estimate may be obtained by extrapolating on observable high frequencies or single-cell information (Moeller *et al.* 2024). Examples of whole-genome single-cell sequencing of hundreds of stem cells in noncancerous tissues exist (Lee-Six *et al.* 2018; Abascal *et al.* 2021; Mitchell *et al.* 2022; Coorens *et al.* 2025), in which the tMB of the samples can be readily extrapolated to larger cell populations (Moeller *et al.* 2024; Mon Père *et al.* 2024). Analogous to the SFS, we link the tMB to the topology of the phylogenetic tree and provide a general formula to compute the expected tMB. We rederive the analytic expression for the tMB at detection from Gunnarsson *et al.* (2021) and provide novel expressions for the tMB after homogeneous treatment response. Eventually, we connect the tMB to the risk of resistance and discuss its relevance for therapeutic interventions.

### Construction of the total mutational burden

Looking at the phylogenetic tree, we denote the number of nodes including leaves with at least one living descendent by $R$, and label them with $i = 1, 2, 3, \ldots, R$. Each node comes with $\mu_i$ unique mutations such that the tMB is

$$B = \sum_{i=1}^{R} \mu_i. \tag{14}$$

Here, $R$ and $\mu_i$ are random numbers and $\mu_i \sim \text{Poiss}(m)$ are independent such that

$$
\begin{aligned}
E[B] &= E[R]E[\mu_1] \\
&= E[R]m
\end{aligned} \tag{15}
$$

and

$$
\begin{aligned}
Var[B] &= Var[\mu_1]E[R] + E^2[\mu_1]Var[R] \\
&= mE[R] + m^2 Var[R].
\end{aligned} \tag{16}
$$

We separate again between newly emerging and preexisting mutations such that the total tMB is the sum of the two contributions, $B^{(tot)} = B^{(new)} + B^{(pre)}$ that are characterized by the following theorem.

**Theorem 2** Suppose the expected total population size over the time interval $[0, t]$ is $N(t')$ with initial size $N_0$, and the probability to grow from size $x$ to $y$ in time $t$ is $p(x \to y, t)$. Then, the expected tMB of newly emerging mutations conditioned on survival and time $t$ is

$$E[B^{(new)}] = \frac{1}{1 - p(N_0 \to 0, t)} \int_0^t 2mbN(t')(1 - p(1 \to 0, t - t'))dt'. \tag{17}$$

Suppose the expected initial SFS at time $t' = 0$ is $E[S_{k'}^{(init)}]$, then the expected tMB of preexisting mutations conditioned on survival and time $t$ is

$$E[B^{(pre)}] = \frac{1}{1 - p(N_0 \to 0, t)} \sum_{k'=1}^{N_0} E[S_{k'}^{(init)}] \, (1 - p(k' \to 0, t)). \tag{18}$$

*Proof.* Supplementary Section H.

The expected tMB can be obtained from Theorem 1 by noting that $E[B] = \sum_k^\infty E[S_k]$. However, the expressions for the expected SFS after treatment are complicated and Theorem 2 then provides more straightforward calculations.

### The total mutational burden at detection

Following Theorem 2, the expected tMB at detection is

$$
\begin{aligned}
E[B^{(det)}] &= 2mN_d \frac{b_1}{d_1}\ln\left(\frac{b_1}{b_1 - d_1} - \frac{N_d^{-1}}{1 - \alpha(t_d)}\frac{d_1}{b_1 - d_1}\right) \\
&\to 2mN_d \frac{b_1}{d_1}\ln\left(\frac{b_1}{b_1 - d_1}\right) \quad \text{for } N_d \to \infty,
\end{aligned} \tag{19}
$$

where $N_d = \tilde{N}(t_d)$ is the expected size at fixed time $t_d$. The expression coincides with the expression for the tMB from Gunnarsson *et al.* (2021), in which the expected tMB was computed by summing over the SFS. We focus on mutations that occur during growth and treatment. Should one be interested in mutations preexisting in the ancestor cell, those will be shared by the entire population and can be added as a constant. Detailed calculations are in Supplementary Section I.

From equation (19), it is apparent that increasing the death rate increases the tMB if measured at the same population size. This is expected, as an increased death rate implies more cell divisions to reach the same population size. However, a considerable fraction of mutations are at low frequencies and will go extinct even if the cell population continues increasing.

In the pure-birth process, the tMB is $E[B^{(det)}] \approx 2mN_d$. To grow from 1 cell to $N_d$ cells, the population undergoes $N_d - 1$ divisions, where each daughter cells obtains a random number of mutations. Thus, the tMB is the sum of $2 \times (N_d - 1)$ independent random variables that are Poisson distributed with mean $m$. Approximating $N_d - 1 \approx N_d$, we have $B^{(det)} \sim \text{Poiss}(2mN_d)$, which

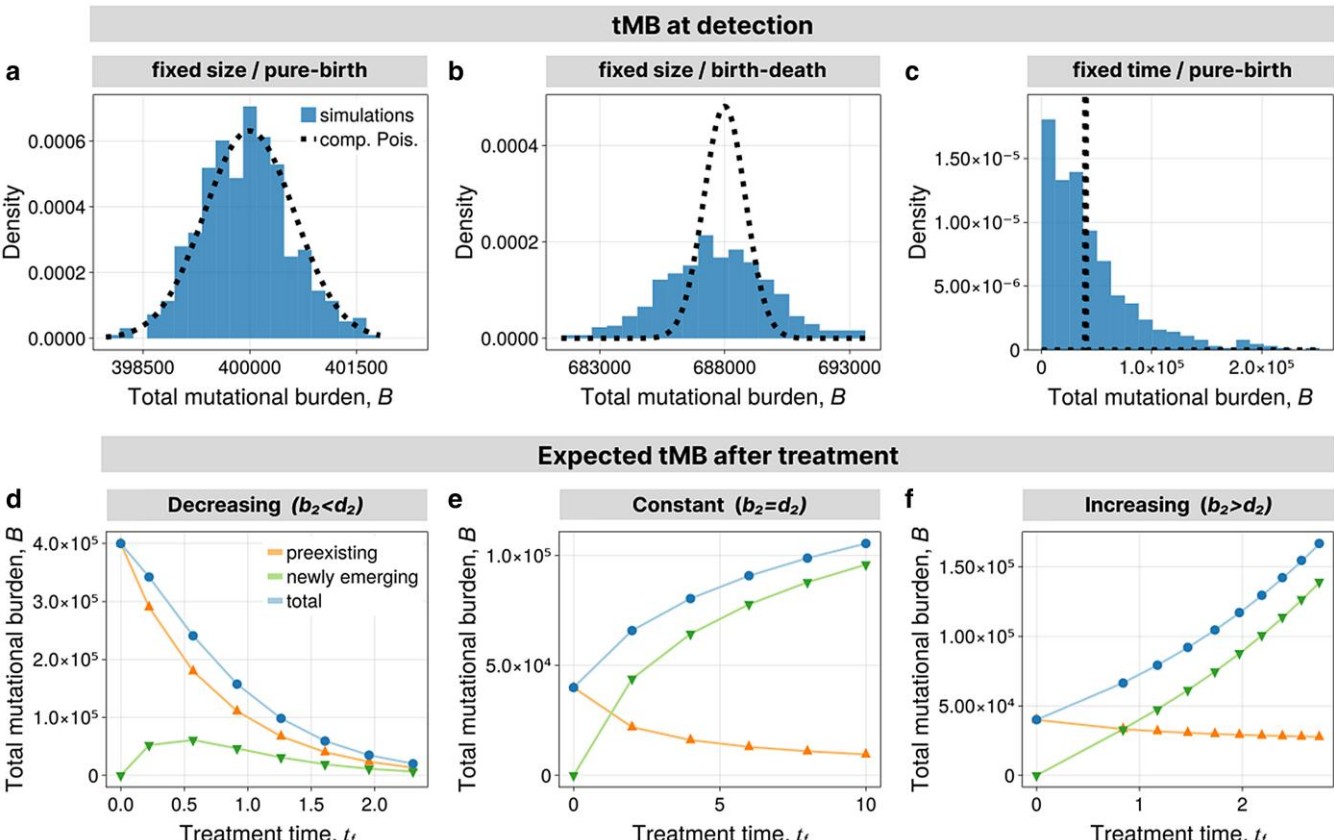

**Fig. 3.** tMB in homogeneous populations. a–c) Distribution of tMB at detection from simulations compared to compound Poisson distribution with mean given by equation (19). a) Pure-birth process conditioned on fixed size. b) Birth–death process conditioned on fixed size. c) Pure-birth process conditioned on fixed time. d–f) Average tMB after treatment from simulations (points and triangles) and predictions (solid lines) using Theorem 2 with explicit formulas in Supplementary Section J. d) For decreasing population. e) For constant population. f) For increasing population. Parameters are in Supplementary Table S2.

agrees with stochastic simulations conditioned on fixed size (Fig. 3a). For observations made on a fixed time, the varying number of cells yields an additional layer of stochasticity. Also, if $d_1 > 0$, there are different growth trajectories to reach size $N_d$. From equation (16), it is clear that in either of the two cases $Var[B^{(det)}] > E[B^{(det)}]$ because the variance in division counts is positive, $Var[R^{(det)}] > 0$, implying an increased genetic heterogeneity between tumors (Fig. 3b and c).

### The total mutational burden after treatment

We compute the expected tMB after homogeneous treatment response using Theorem 2. We assume initial size $N'_d$, expected SFS $E[S_k^{(init)}] = E[S_k^{(det)}]$ approximated by equation (9), and treatment interval $[0, t_f]$. The tMB of mutations emerging during treatment is solved exactly. For decreasing or increasing population (i.e. $b_2 \neq d_2$), it is

$$E[B^{(new)}] = 2mN_f \frac{b_2}{d_2} \ln\left(\frac{b_2}{b_2 - d_2} - \frac{N'_d/N_f}{1 - \alpha(t_f)} \frac{d_2}{b_2 - d_2}\right) \quad (20)$$

that simplifies to equation (19) for $N'_d = 1$. For the constant population ($b_2 = d_2$), it is

$$E[B^{(new)}] = 2mN_f \ln(1 + b_2 t_f). \quad (21)$$

The expression for mutations preexisting to treatment is more involved due to summation over the initial SFS. However, we find a simple approximation for $d_1 = 0$, which we heuristically extend for general $d_1 > 0$ that reads

$$E[B^{(pre)}] \approx E[B^{(det)}] \frac{1 - \alpha(t_f)}{\alpha(t_f)} \log\left(\frac{1}{1 - \alpha(t_f)}\right). \quad (22)$$

The separation between $b_2 = d_2$ and $b_2 \neq d_2$ is made through $\alpha(t_f)$ defined in equation (1). The approximation performs very well in all three cases with small deviations for increasing $d_1$, in which case the approximation yields an useful upper bound (Supplementary Fig. S3).

In Fig. 3d–f, we show the alignment of the exact theoretical predictions with stochastic simulations. It is worth noting that most mutations contributing to tMB are at low frequencies. After exponential growth, it is estimated that 50% of unique mutations occur in only 1 cell (Gunnarsson et al. 2021). After treatment, in particular, if the population remains approximately constant, this asymmetry is less extreme (Moeller et al. 2024). Still, the large majority of mutations remain at low frequencies.

### Relationship to genetic resistance against therapy

The tMB is linked to the risk of resistance. An increasing number of mutations comes with an increased chance of genetic resistance mechanisms, which is confirmed in clinical data for various cancer types and treatments (Andor et al. 2016; Turajlic et al. 2019). Mathematically, the tMB is connected to the probability of having a resistant subpopulation through a probability $p_r$ that a specific

mutation leads to resistance. Given the tMB is $B(t)$ at time $t$ the probability that none of those has led to resistance is $\Pr(\text{sensitive}) = (1 - p_r)^{B(t)}$. Although this picture is oversimplified, it makes clear that reducing the tMB generally decreases the risk of resistance. Notable exceptions are immunotherapies, which rely on the accumulation of neoantigens to be effective. In this case, an increased tMB is a double-edged sword: Higher tMB comes with an increased treatment efficiency due to a higher load of neoantigens necessary for the immune system to detect the cancer cells as invaders but also increases the chance of resistance mechanisms such as immune escape (Rizvi *et al.* 2015; McGranahan *et al.* 2016; Lakatos *et al.* 2020).

In Fig. 3d–f, we show that mutations arising during treatment contribute significantly and often dominate the tMB shortly after treatment initiation. Especially in the constant population scenario, over time, preexisting mutations are almost fully depleted and most mutations will have arisen during treatment. This is an important observation, especially for the planning of possible adaptive clinical trials that are already ongoing or in preparation (Zhang *et al.* 2017; Mukherjee *et al.* 2024). A constant population during treatment approximates the population dynamics in adaptive therapies. It is clear that drugs suppressing the accumulation of new mutations—if feasible—would significantly reduce the risk of novel acquired genetic resistance. Our analysis supports and quantifies this intuition, and furthermore suggests that at the same time preexisting genetic diversity is quickly depleted.

There also has been a long debate if treatment resistance is mostly caused by selection on preexisting genetic diversity or resistance is acquired during periods of treatment (Wang *et al.* 2019). These questions are not only related to cancer therapies but similarly can be asked for antibiotic or antiviral therapies (Blair *et al.* 2015; Irwin *et al.* 2016). Given the connection between the tMB and the probability of resistance, the results in Fig 3d–f suggest that the dominant mechanism depends on the treatment mechanism, duration, and the resulting population dynamics of the treated population. In case of a shrinking population, diversity is for the most part dominated by preexisting mutations. In contrast, in constant or growing populations, genetic diversity is quickly dominated by newly acquired mutations. On this end, our study complements the theoretical literature on treatment resistance (Komarova and Wodarz 2005; Michor *et al.* 2006; Bozic *et al.* 2013; Patil *et al.* 2024) with the potential to guide future planning of combination or adaptive cancer therapies and to assess the underlying mechanisms of failed clinical trials.

## The single cell mutational burden distribution

The scMB distribution has started recently to attract attention. Whole genome scMB information across different normal human tissues and ages have become available (Abascal *et al.* 2021; Mitchell *et al.* 2022; Coorens *et al.* 2025) and theoretical models of scMBs during homeostasis have shown to complement information encoded in the SFS (Moeller *et al.* 2024). We extend on these considerations and discuss the scMB distribution and its properties under scenarios of cancer growth and during treatment focusing on homogeneous populations.

### Construction of the single cell mutational burden distribution

The scMB distribution $M_j$ is linked to the phylogenetic tree through the divisional distributions, which counts the number of cells $D_l$ with $l$ divisions. In terms of the tree topology, $l$ is the leaf depth that is the distance, i.e. number of nodes, from the leaf to the root (Fig. 1a). The distribution $\{D_l\}_{l=1}^{\infty}$ partitions the cell population

into classes of cells according the number of divisions. Picking a cell with label $n'$ that has undergone $l$ divisions, labeling its divisions by $n = 1, 2, \ldots, l$ and noting that on each division the cell accumulated $\mu_{n',n}$ mutations, we can write the scMB distribution in term of the divisional distribution. Namely, we have

$$M_j = \sum_{l=1}^{\infty} \sum_{n'=1}^{D_l} 1_{\left\{ \sum_{n=1}^{l} \mu_{n',n} = j \right\}}, \tag{23}$$

where $1_{\{x=y\}}$ is an indicator function that is 1 if $x = y$ and 0 otherwise. The following theorem characterizes the expected scMB distribution.

**Theorem 3** Given population size $N$, the expected scMB distribution is expressed in terms of the expected divisional distribution $E[D_l]$ by

$$E[M_j] = \sum_{l=1}^{\infty} E[D_l] \frac{(lm)^j e^{-lm}}{j!}. \tag{24}$$

Starting with $N_0$ cells that have undergone zero divisions, the mean-field solution for the expected divisional distribution for a population of size $N$ at time $t$ is

$$E[D_l] \stackrel{\text{mf}}{=} N \frac{(2bt)^l}{l!} e^{-2bt}, \tag{25}$$

such that the probability mass $P_l = \frac{E[D_l]}{N}$ follows a Poisson distribution with mean $2bt$.

*Proof.* Supplementary Section K.

For exponentially growing populations, equation (24) was recently reported by Morison *et al.* (2023), and equation (25) was derived by Kharlamov (1969) and Williams *et al.* (2018). Furthermore, it has been shown that the divisional distribution is normal distributed as time goes to infinity (Bühler 1971; Samuels 1971; Bühler 1972) and some exact expressions with higher complexity were obtained (Steel and McKenzie 2001; Cheek and Johnston 2023). Since the mean-field solution shows good agreement with simulations up to a small correction for small population size, we proceed with the much simpler mean-field solution.

The average number of divisions for a single cell underlies an inspection bias (Cheek and Johnston 2023). Consider a population that has lived for time $t$ with birth rate $b$. Picking a leaf of the phylogenetic tree at random and tracing it back to the root, one may naively estimate that the leaf has undergone $bt$ divisions on average. Yet, equation (25) predicts an average division number of $2bt$. To understand the increased number of divisions, note that some cells proliferate more than other by chance. Cells with more divisions have a larger number of offspring and contribute more to the population average.

### The single cell mutational burden distribution at detection

Following Theorem 3, we first obtain the divisional distribution from equation (25) and then the scMB distribution from equation (24) for a cell populations starting with 1 cell growing to $N_d$ cells. A comparison between the analytic prediction and simulations revealed a small constant offset of the mean. To resolve this discrepancy, we looked at the average number of divisions per cell defined by $\bar{l} = \frac{1}{N_d} \sum D_l$. Rigor treatment of the pure-birth process

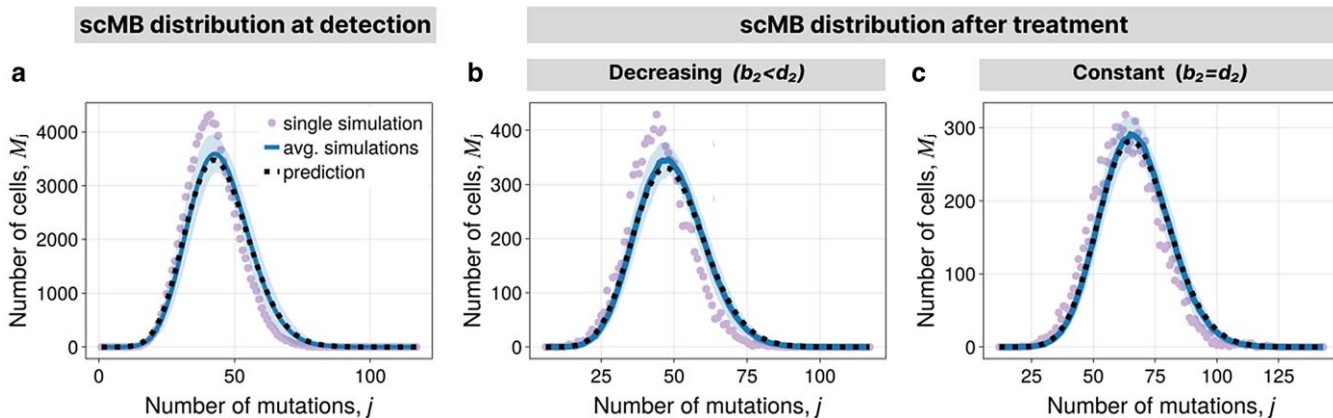

**Fig. 4.** scMB distribution in homogeneous populations. a) scMB distribution at detection for the pure-birth process. Average over many simulations with the 1-sigma interval, single realization, and the prediction from Theorem 3 with divisional distribution as in equation (26). b and c) scMB distribution after treatment. Average over many simulations with the 1-sigma interval, single simulation, and the prediction from Theorem 3 with divisional distribution as in equation (27). b) For decreasing population. c) For constant population. Parameters are in Supplementary Table S3.

shows that the expected average number of divisions is given by $E[\bar{l}^{(\text{exact})}] = 2(H_{N_d} - 1)$. From this expression, it follows that the mean-field solution obtained through equation (25) differs from the exact solution by a constant 2 when conditioned on fixed size or $\ln(4)$ when conditioned on fixed time (Supplementary Section L). Focusing on fixed-size observations, we integrate the correction constant into the divisional distribution setting

$$P_l \sim \text{Poiss}(2bt_d - 2), \qquad (26)$$

where $t_d$ is defined as the mean time to reach size $N_d$ for the first time (see Supplementary Section B). The adjusted scMB distribution shows excellent agreement with simulations of the pure-birth process (Fig. 4a), and good but not perfect agreement when deaths are included (Supplementary Fig. S5). The scMB distribution of single realization has the same shape as the expected distribution, although it comes with a small random shift of the mean. These shifts are caused by stochastic effects at small population sizes leading to mutations shared by a large fraction of the population.

### The single cell mutational burden distribution after treatment

For homogeneous treatment response, the analytic predictions for decreasing, constant, and increasing populations under treatment are the same. As long as the final population size is large, i.e. $N_f \gg 1$, small population effects can be neglected such that the average number of divisions obtained during treatment time $t_f$ is $E[\bar{l}] = 2bt_f$. Those divisions are again Poisson distributed and together with divisions before treatment, we have

$$P_l \sim \text{Poiss}\big((2bt_d - 2) + 2bt_f\big), \qquad (27)$$

where $t_f \approx \frac{\ln(N_f/N_d)}{b_2 - d_2}$. We put $E[D_l] = N_f P_l$ into equation (24) to obtain the scMB distribution showing good alignment with simulations (Fig. 4b, c and Supplementary Fig. S6).

The average scMB increases linear in time (Supplementary Fig. S4), in concordance with experimental observations of healthy human tissues (Abascal et al. 2021). Whereas this is also true for some limiting cases for the tMB normalized over the population size, the tMB shows, in general, more complex nonlinear behavior (Fig. 3d–f). Similarly to the observable SFS, the scMB distribution

does not change its shape after homogeneous treatment response yielding a null model to test homogeneity. In contrast to the SFS, the scMB distribution differs from the null hypothesis only for heterogeneity in birth rates, i.e. resistant cells have different birth rates than sensitive cells. The scMB distribution relies on high-resolution single-cell sequencing data such as those from Mitchell et al. (2022). Like the SFS, the scMB distribution must be corrected for potential copy number alterations. Once the scMB is retrieved from appropriate data, the scMB distributions allows decoupling mutation rate and birth rate by studying its variance (Moeller et al. 2024).

## Genetic intra-tumor heterogeneity upon emerging resistance

So far, we discussed properties of gITH in cell populations with homogeneous treatment response. For certain treatment regimes such as chemo or radiotherapy in early stage disease, this seems a reasonable scenario. However, in other cases, the emergence of resistant subpopulations is a major problem. Genetic resistance is well documented in many targeted cancer therapies, where specific point mutations induce resistance (Diaz Jr et al. 2012; Khan et al. 2018). Resistance mutations give rise to resistant clones leading to a cell population, in which some cells are less or not affected by treatment (Fig. 1c). Building upon the results for homogeneous populations, we first study the size distribution for resistant clones, followed by investigations of the SFS and scMB distribution after heterogeneous treatment response.

### The resistant clone size distribution

We distinguish between mutations without effect accumulated at rate $m$ and mutations that cause resistance accumulated at rate $\nu$ per cell and division. We denote the number of resistant clones of size $\kappa$ by $\Theta_\kappa$ and call $\{\Theta_\kappa\}_{\kappa=1}^N$ the CSD. In our setup, the CSD and the SFS have a conceptional difference: the CSD is not nested, i.e. clones are not overlapping, whereas the SFS allows nested lineages. However, since $\nu$ is very small, resistance mutations have negligible effect on the growth dynamics of sensitive cells. As a consequence, we can approximate the CSD by adapting Theorem 1 (Supplementary Section M). Then, at detection, the expected CSD is

$$E[\Theta_\kappa^{(\text{det})}] \approx \frac{1}{1 - p(1 \to 0,\, t_{\text{d}})} \int_0^{t_{\text{d}}} 2\nu e^{(b_{s,1} - d_{s,1})t} p(1 \to \kappa,\, t_{\text{d}} - t)\, dt, \quad (28)$$

where the term for conditioning on survival $\frac{1}{1 - p(1 \to 0,\, t_{\text{d}})}$ is parameterized with growth parameters for sensitive cells $b_{s,1}$, $d_{s,1}$ while the term $p(1 \to \kappa,\, t_{\text{d}} - t)$ is parameterized with growth parameters for resistant cells $b_{r,1}$, $d_{r,1}$. Assuming that the growth parameters of sensitive and resistant cells are the same in the absence of treatment, the CSD takes the same form as the SFS in homogeneous populations in equation (9). Focusing on the limit $N_{\text{d}} \to \infty$ and writing $b_1 := b_{s,1} = b_{r,1}$ and $d_1 := d_{s,1} = d_{r,1}$, we have

$$E[\Theta_\kappa^{(\text{det})}] \approx 2\nu N_{\text{d}} \sum_{l=0}^{\infty} \frac{\left(\frac{d_1}{b_1}\right)^l}{(\kappa + l)(\kappa + l + 1)} \quad (29)$$

This is not surprising, given above assumptions, resistance inferring mutations are neutral before treatment.

Next, we consider the CSD after treatment time $t_f$. As initial condition, we assume fixed size $N_d'$, and initial CSD $E[\Theta_\kappa^{(\text{init})}] = E[\Theta_\kappa^{(\text{det})}]$. We adapt Theorem 1 and obtain the expected CSD of clones that emerged before and after detection that read

$$E[\Theta_\kappa^{(\text{new})}] \approx \int_0^{t_f} 2\nu b_{s,2} N_d' e^{(b_{s,2} - d_{s,2})t'} p(1 \to \kappa,\, t_f - t')\, dt' \quad (30)$$

and

$$E[\Theta_\kappa^{(\text{pre})}] \approx \sum_{\kappa'=1}^{N_d'} E[\Theta_{\kappa'}^{(\text{det})}] p(\kappa' \to \kappa,\, t_f). \quad (31)$$

In equation (30), the relevant mutation generation is caused by sensitive cells ($2\nu b_{s,2} N_d' e^{(b_{s,2} - d_{s,2})t'}$). In both equations (30) and (31), the growth of clone sizes $p(a \to n, t)$ is parameterized for resistance cells $b_{r,2}$ and $d_{r,2}$.

## Subdominance of the largest resistant clone

In an exponentially growing population, the first mutant clone has a large advantage as all later arriving clones need to catch up to become dominant. We thus asked whether the first resistant clone always dominates all other later arising resistant clones. Here, we show that this is not the case. We focus on detection since a dominating resistant clone at detection also dominates after treatment.

First, we find that the first arriving clone is not always the largest. We derived the probability densities for the arrival time $T_1$ of the first clone and the second clone $T_2$ given that $T_1 = t_1$ and survival of drift (Supplementary Section N). The long-term growth of the clones can be written as $V_1 e^{(b_1 - d_1)(t_d - T_1)}$ and $V_2 e^{(b_1 - d_1)(t_d - T_2)}$, where $V_1$ and $V_2$ are exponentially distributed with mean $\frac{b_1 - d_1}{b_1}$ (Theorem 1 from Durrett 2015). The difference in their size at time $t_d$ is then

$$V_1 e^{(b_1 - d_1)(t_d - T_1)} - V_2 e^{(b_1 - d_1)(t_d - T_2)}, \quad (32)$$

which we sample (see Materials and Methods) and illustrate in Fig. 5a. In roughly one-third of the cases, the size difference is in the negative range (illustrated in red). This means that the second clone can outgrow the first clone purely by drift. We obtained similar results by assessing the output of our simulation framework (Supplementary Table S5). Implementing a resistance cost, we observe that instances in which the second clone outgrows the first become more frequent and that the clone sizes are more similar

to each other. These effects would become even more pronounced for possible fitness distributions of resistant clones.

Next, we considered the largest clone at time $t_d$ instead of the first arriving clone. Interpreting the CSD as a probability distribution for clone sizes, we can use methods from order statistics (David and Nagaraja 2004) and find an approximate probability density for the size of the largest resistant clone $\tilde{\kappa}$ that reads

$$f(\tilde{\kappa}) = \frac{A}{\tilde{\kappa}^2} \left(1 - \frac{1}{\tilde{\kappa}}\right)^{A-1} \quad \text{with } A = \frac{b_1}{b_1 - d_1} 2\nu N_d \quad (33)$$

and is valid for $\nu N_d \gg 1$. The derivation is presented in Supplementary Section N and we validated our approximation with simulations (Fig. 5b). Notably, Cheek and Antal (2018) derived an expression for the largest clone in the limit of large populations and small mutation rates, which yields a similar result (Supplementary Section N).

We compare the size of the largest resistant clone to the total number of resistant cells. Given small mutation rate $\nu$ but large population sizes $N_d$ and $\nu N_d \gg 1$, the probability of having $R$ resistant cells at the time of detection is described by a Landau distribution (Kessler and Levine 2013, 2015). We have

$$P(R) = \frac{b_1 - d_1}{b_1} \frac{1}{2\nu N_d} P_{\text{Landau}}\left(\frac{b_1 - d_1}{b_1} \frac{R}{2\nu N_d} - \ln(2\nu N_d)\right) \quad (34)$$

that shows good agreement with simulations but with a small offset (Fig. 5b).

Both distributions have a fat tail that decreases according to a power law with exponent 2. The mean and the median of the largest clones $\tilde{\kappa}$ are below the mean and median of $R$. Interestingly, the median fraction of all resistant cells is weakly increasing with detection size $N_d$ and is predicted to be around $3.0 \times 10^{-5}$ for biologically feasible parameter values. In contrast, the median size of the largest resistant clone is approximately constant for different detection sizes and is around $4.0 \times 10^{-6}$ (Fig. 5c). This is reasonable as newly resistant cells continue to emerge from sensitive cells during growth and thus increase the fraction of resistant cells as the tumor grows. We conclude that the largest clone may make up a significant fraction but probably not a dominating fraction of the resistant subpopulation.

This conclusion is consistent with previous studies of Bozic and Nowak (2014) who derived the size distributions of individual clones ordered by appearance. Specifically, the authors showed that the median size of first arriving clone is $\frac{1}{\sqrt{2}-1} \approx 2.41$ times larger than the median size of the second arriving clone. Through simulations, we find similar values in the case of neutral growth dynamics, while the median sizes become more similar in the case of costly resistance (Supplementary Table S5).

## The site frequency spectrum

We assume resistant cells to be neutral in the absence of treatment such that the expected SFS at detection is unchanged given by equation (9). In the following, we focus on the SFS after treatment. We separate between neutral mutations that emerged in sensitive cells $S_k^S$ and mutations that emerged in resistant cells $S_k^R$. We label $K$ resistant clones with $i = 1, 2, \ldots, K$ such that

$$S_k = S_k^S + S_k^R \quad \text{with } S_k^R = \sum_i^K S_k^{R,i}. \quad (35)$$

By definition, $S_k^S$ combines mutations in sensitive cells and mutations in resistant cells acquired in sensitive ancestors.

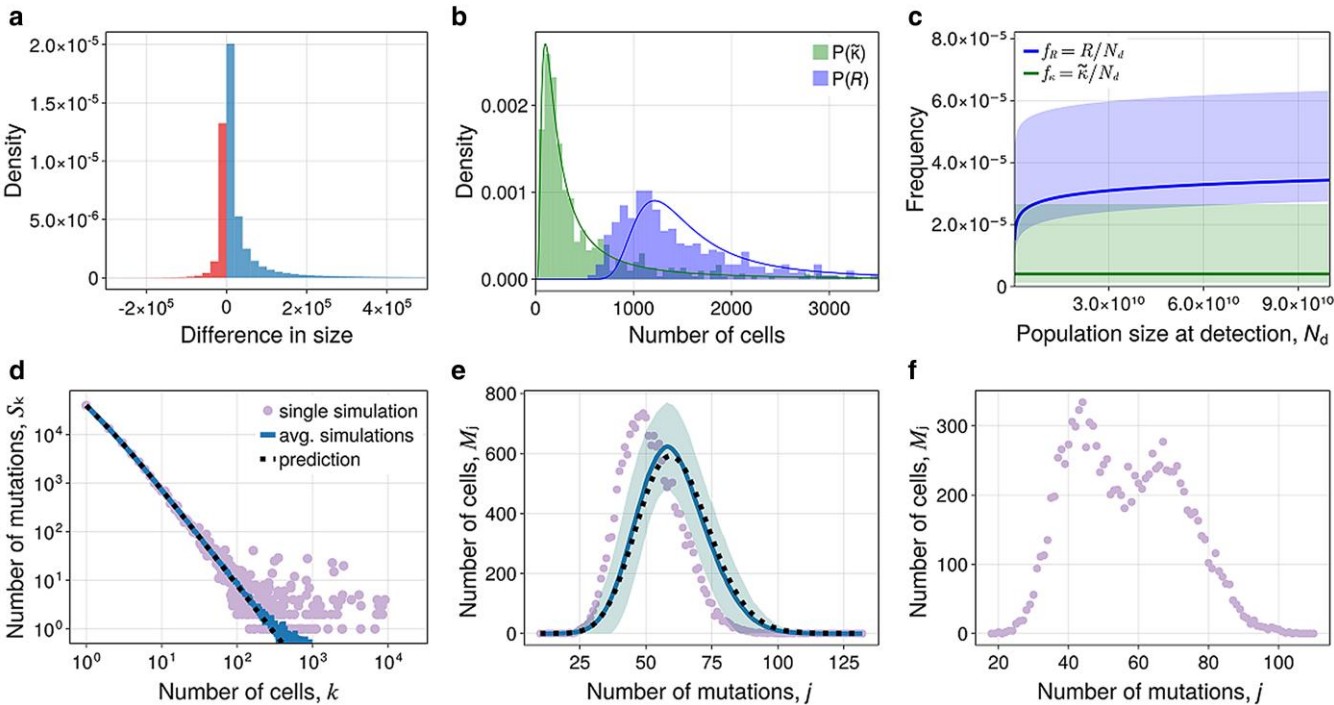

**Fig. 5.** gITH in heterogeneous populations. a) Difference in size between the first and second resistant clone at detection given by equation (32). Negative differences are shown in red, and positive differences in blue. b) Size distributions for the largest resistant clone and the resistant subpopulation at the time of detection. Simulations as histograms, and predictions as solid lines given by equations (33) and (34). c) Distribution of frequencies for the largest resistant clone and the resistant subpopulation at the time of detection. The median frequency is shown as a solid line with shadow ranging from the 10% to the 90% quantiles. d) SFS after treatment upon emergence of resistance from simulations and prediction of the compound tail in equation (36). Parameters are set such that all sensitive cells are extinct. e) scMB distribution upon emergence of resistance in which birth rates remain unaffected. For the prediction, we used treatment time $t_r$ measured from one simulation. f) scMB distribution for a single population upon emergence of resistance with change of birth rates. Parameters are in Supplementary Table S4.

If resistance mutations lead to full resistance, i.e. $b_r := b_{r,1} = b_{r,2}$ and $d_r := d_{r,1} = d_{r,2}$, then the compound SFS of all resistant clones takes again the known form

$$E[S_k^R] = \sum_{\kappa=1}^{N_f} E[\Theta_\kappa] E[S_k(\kappa)] \approx 2mN_r \sum_{l=0}^{\infty} \frac{\left(\frac{d_r}{b_r}\right)^l}{(k+l)(k+l+1)}, \quad (36)$$

where $N_r$ is the number of resistant cells, $E[S_k(\kappa)]$ is the SFS of an exponentially growing cell population at size $\kappa$. The calculations are presented in Supplementary Section O and alignment with simulations is shown in Fig. 5d.

To make progress on $S_k^S$, we assume sufficiently long treatment times such that all sensitive cells go extinct. Then, $S_k^S$ consists of mutations in resistant cells that are clonal in at least one resistant clone. If we denote $\vec{\kappa} = (\kappa_1, \kappa_2, \ldots, \kappa_K)$ as a vector of clone sizes $\kappa_i$ with indices $i \in X$, we have

$$S_k^{S,R} = \sum_{Y \in 2^X} C_Y \delta\left(\sum_{j \in Y} \kappa_j = k\right). \quad (37)$$

Here, the first sum goes over all possible combinations Y in the set of indices X that is the power set $2^X$, and $C_Y$ is the number of mutations shared between the chosen clones. Those mutations will arise as peaks in the SFS (Fig. 5d and Supplementary Fig. S7). The heights and the locations of the peaks carry information on the timing and selection of resistant clones. However, a more in-depth understanding of the peaks requires analyzing the

relatedness between clones and goes beyond the scope of this study.

Our results of the SFS upon emergence of resistance are restricted to a number of assumptions (see Discussion). However, the results showcase how the SFS after heterogeneous treatment response differs from the homogeneous null. Next to our study, we are only aware of Bonnet and Leman (2024) who analyzed the SFS in this scenario. While Bonnet and Leman (2024) provide more mathematical rigor, their results are confined to mutation accumulation after detection.

### The single cell mutational burden distribution

From our previous investigations, we know that the scMB is not affected by the death rate but only the birth rate. Thus, if the birth rate is unaffected in the selection process then the scMB distribution remains the shape described in Theorem 3, which we confirmed with simulations (Fig. 5e). However, if the birth rate of the resistant cell lines differs from the sensitive cells, then the scMB distribution will differ too (Fig. 5f).

We write the scMB distribution of the resistant population as $M_j^R = \sum_{i=1}^K M_j^{R,i}$, where $M_j^{R,i}$ is the scMB distribution of an individual clone with label $i$. Together with the sensitive cells, the total expected scMB distribution is

$$\begin{aligned} E[M_j] &= E[M_j^S] + E[M_j^R] \\ &= E[M_j^S] + \sum_{\kappa=1}^{N_f} E[\Theta_\kappa] E[M_j(\kappa)]. \end{aligned} \quad (38)$$

The total scMB distribution consists of overlapping scMB distribution of sensitive cells and resistant clones. If the birth rate changes in resistant cells during treatment but is the same among all resistant cells, the majority of resistant cells consist of clones that arose before treatment with the same expected scMB leading to a combined single peak of resistant cells next to the peak of sensitive cells (Fig. 5f). If the birth rate of resistant clones changes already before treatment, the clones will have different expected scMB leading to many overlying distributions.

## Discussion

We used stochastic processes to investigate expected patterns of genetic heterogeneity in cancer before and after treatment. We discussed three summary statistics of gITH (SFS, tMB, and scMB) for different treatment scenarios. For homogeneous treatment response, we found that the SFS keeps its $k^{-2}$ power-law tail regardless whether the cancer is shrinking, approximately constant or growing, and the scMB distribution is described by two entangled Poisson processes. Those predictions build the basis for possible homogeneity tests, and can provide insights into the mechanisms underlying resistance evolution. Similar usage of a null hypothesis of the SFS has led to biological insights into selection processes of healthy tissues (Schenck *et al.* 2022) and cancer before treatment (Williams *et al.* 2016). We further characterized the nonlinear behavior of the tMB separating between preexisting and newly emerging mutations. The tMB relates to the probability of having resistant cells and the scMB may be interpreted as scMB of antigens important for the response to immunotherapies (Lakatos *et al.* 2020; Zapata *et al.* 2023).

Next, we investigated the effect of treatment resistance on gITH assuming a simple model of genetic resistance reasonable for treatment of late-stage cancers with targeted therapies that often show resistance due to specific point mutations (Diaz Jr *et al.* 2012; Khan *et al.* 2018). We found that the resistant subpopulation will consist of many resistant clones that arose from sensitive cells, and the largest resistant clone likely makes up a considerable but not dominating fraction of the total resistant subpopulation, which is concordant with previous theoretical studies (Bozic and Nowak 2014; Bozic *et al.* 2016). We showed that resistant clones leave traces in the SFS, and change the scMB distribution only if birth rates are affected.

Data generation often comes with additional complexities that must be incorporated before our results can be readily compared to the data. Sequencing comes with sampling biases and sequencing noise. Next to mutations accumulated during tumor development, further artifacts are added during sample preparation, sequencing and mapping errors restricting our ability to confidently assess low-allelic frequencies. A sequencing depth of 100× is suggested to detect variants present in 10% of the sample and a sequencing depth of 1,000× is suggested to detect variants present in 1% (Stead *et al.* 2013; Tanner *et al.* 2021). This threshold will likely be lowered in the future with technological advances such as Duplex sequencing (Kennedy *et al.* 2014). Our homogeneity test requires that resistant subclones exceed these thresholds with a detectable mutational distance from the sensitive wildtype clone. Another restriction emerges as not the entire cancer cell population but only a sample is sequenced. In a well-mixed population, the SFS in a sample of $n$ cells is mathematically obtained through binomial sampling of the $N$ cells in the population. In this case, the characteristic power-law tail remains in the sampled SFS (Theorem 3 from Durrett 2015) such that testing $S_k^{(tot)} \sim k^{-2}$ is indeed a valid homogeneity test. More problematic

are correlations in the sampling particularly due to spatial structure. For solid tumors, the risk is high that a single sample contains only cells from a single clone although the tumor itself is multiclonal. The chance of missing out multiclonality can be reduced by multiregion sequencing (Gerlinger *et al.* 2012; Frankell *et al.* 2023).

For generality and tractability, here we used a minimal model when defining our theoretical framework. Our model assumes well-mixed populations with no competition. However, solid tumors grow under the influence of spatial constraints. Spatial constraints increase local competition and can profoundly impact evolutionary dynamics (West *et al.* 2021; Noble *et al.* 2022; Stein *et al.* 2024) which in return can impact the development of treatment resistance (Bacevic *et al.* 2017). We employ a simplistic model for treatment in which we change growth parameters with the application of treatment. We did not consider that treatment can increase mutation rates, which is observed, for example, in radiotherapies or platinum-based drugs (Pich *et al.* 2019; Kocakavuk *et al.* 2021). Our treatment model does not include possible delays in treatment response, limited drug delivery due to physical barriers or limitations due to treatment toxicity, which are often studied under the umbrella of pharmacokinetics and pharmacodynamics (Dayneka *et al.* 1993; Upton and Mould 2014). Lastly, we did not consider interactions with the tumor microenvironment or other nongenetic mechanisms, which can impact the evolution of treatment resistance (Marine *et al.* 2020; Nam *et al.* 2021).

Investigating the cost of resistance in more depth as well as the impact of varying treatment responses for different resistant clones will be of interest in future studies. Our models are suitable for adding additional complexities, e.g. time-varying or adaptive treatment regimes and their impact on gITH is very much a question of interest (Gatenby *et al.* 2009; Zhang *et al.* 2017; Viossat and Noble 2021). We hope that our results will set the stage for further theoretical and data-driven investigations to unravel the patterns of gITH after treatment.

## Materials and methods

Computer simulations were implemented in the Julia programming language and are based on the Gillespie algorithm (Gillespie 1976, 1977). We adapted the simulation framework developed by Yu *et al.* (2024). The cell population is saved in a tree structure. The root of the tree represents the ancestor cell and the leaves represent living cells. In case of a death event the chosen leaf is removed from the tree. In case of a birth event two new leaves are created and each carries the number of mutations drawn from a Poisson distribution. Then, each node that is not a leaf represents an ancestor of a currently living cell.

To obtain the SFS, we iterate over all nodes and count the number of mutations unique to this node and its ancestors. If the node with label $i$ has $\mu_i$ mutations and leads to $k$ leaves, we increase $S_k$ by $\mu_i$. After iterating over all nodes, we have the SFS.

For the scMB distribution, we iterate over all leaves. On top of the mutation of the leaf itself, we add its parent's mutations, then its parent's mutations and go on until we reach the ancestor cell. The final sum is the scMB of this leaf. After iteration over all leaves, we have the scMB distribution.

Numerical solutions of integrals were computed using the HCubature module in the Julia programming language (Johnson 2017) that is based on an adaptive algorithm (Genz and Malik 1980). Integrals were computed up to a relative tolerance of $10^{-3}$.

Mathematical analysis was supported using *Wolfram Mathematica*.

We balance efficiency and accuracy for the computation of the probability mass function of the birth–death process $p(a \to n, t)$. We always use exact solutions for the pure-birth process, pure-death process or when the population start from a single cell (Bailey 1964). Otherwise, we use exact solutions written as computationally stable sum (Tavaré 2018) for $\min(a, n) \leq 50$ and use the (renormalized) saddlepoint-approximation method solution for $\min(a, n) > 50$ (Davison *et al.* 2021). Details of the approximations are described in Supplementary Section A.

Random samples for random variables $T_1$ and $T_2$ were generated using inverse transformation sampling (Devroye 2006). Therefore, a random number $u$ is drawn from a uniform distribution in (0, 1) and is then transformed into a random number $x$ with cumulative density function $F_X(x)$ by plugging the uniform random number into the inverse distribution function $x = F_X^{-1}(u)$.

We fit the exponent $\gamma$ of the power-law $S_k \sim k^\gamma$ to analyze the SFS obtained from simulated data. Therefore, we use a maximum likelihood estimator for discrete power-law distributions from Clauset *et al.* (2009). Adapted for our purposes, we have

$$\hat{\gamma} = 1 + B_{k_{\min}} \left( \sum_{k=k_{\min}}^{N} S_k \ln \frac{k}{k_{\min} - \frac{1}{2}} \right), \tag{39}$$

where $k_{\min}$ defines the smallest site frequency used for the fitting, $B_{k_{\min}} = \sum_{k=k_{\min}}^{N} S_k$ is the number of mutations that occur in at least $k_{\min}$ mutations and $N$ is the population size.

Parameters for Figs. 2, 3, 4, and 5 are found in Supplementary Tables S1, S2, S3, and S4 respectively.

## Data availability

For the simulations, we adapted a *Julia* package that was developed by Yu *et al.* (2024). The original package is available at https://github.com/jessierenton/SomaticEvolution.jl. The adapted package is available at https://github.com/alexsteininfo/TreeStatistics.jl and code to run and analyze the simulations is available at https://github.com/alexsteininfo/GITH-Treatment-Patterns.

Supplemental material available at GENETICS online.

## Acknowledgments

We thank Nathaniel Mon Père, Christo Morison, two anonymous reviewers, and the editor, Guillaume Martin, for helpful comments at different stages of the manuscript.

## Funding

A.S. was supported by the European Union's Horizon 2020 research and innovation program under the Marie Skłodowska-Curie grant agreement No. 955708. B.W. was supported by a Barts Charity Lectureship (grant no. MGU045) and a UKRI Future Leaders Fellowship (grant no. MR/V02342X/1).

## Conflicts of interest

The author(s) declare no conflicts of interest.

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

*Editor: G. Martin*