## [Peer Review File · Genetics]

On the patterns of genetic intra-tumour heterogeneity before and after treatment

Alexander Stein and Benjamin Werner

NOTE: The reviews and decision letters are unedited and appear as submitted by the reviewers.

In extremely rare instances and as determined by a Senior Editor or the EIC, portions of a review may be redacted. If a review is signed, the reviewer has agreed to no longer remain anonymous.

The review history appears in chronological order.

Review Timeline:

Submission Date:	2024-11-12
Editorial Decision:	2025-01-10
Resubmission Received:	2025-03-27
Accepted:	2025-05-13

January 10, 2025

GENETICS-2024-307634

On the patterns of genetic intra-tumour heterogeneity before and after treatment

Dear Dr. Stein:

Two experts in the field have reviewed your manuscript, and I have read it as well. The manuscript tackles a timely question, that is of interest beyond cancer biology, and the main figures show clear tests of the analytical predictions, which is appreciable. The structure with all proofs in appendix is a good strategy given the focus of genetics. Therefore, while your manuscript is not currently acceptable for publication in GENETICS, we would welcome a substantially revised manuscript. Both reviewers have comments and concerns to be addressed in a revised manuscript. You can read their reviews at the end of this email.

Both reviewers point to important aspects that should all be tackled before resubmission. In particular, the links to existing theoretical results should be made clearer, and the biological limits of the model's simplifications in applying to actual sequence data in cancer. I would add to that second point that pharmacokinetics and pharmacodynamics may affect the outcomes compared to a simple fixed birth and death rate before/after treatment (which would find its place in the part dedicated to limits 1.491-500, a part that requires some extensions, see rev#1 & 2).

The introduction is fairly short and could be expanded (reasonably) with references to existing work on these questions, possibly to reviews in the case of large bodies of literature, rev#2 has provided several suggestions. As ref 16 provides the SFS for $N_0=1$, it should be stated from the onset that the results on SFS are extensions of this ref when $N_0 > 1$ (homogeneous), and when two clones with varying rates are present (resistant + sensitive). The same goes a priori for the tMB, where the link to existing result seems partially lacking (see rev#2). When dealing with standing variance in an expanding asexual (pre-treatment), reference to the fluctuation test analysis literature should be done (from the intro) as it has 80 years of theoretical development since the Lea Coulson treatment (of what I believe is the same process: I may be wrong, but then distinction should be made with this important body of literature). This literature includes models of the problem when the mutations carry a cost before treatment (see rev#1's comment); this is sometimes called the Mandelbrot-Koch model eg (Mandelbrot 1974, Koch 1982) and has a long history.

Upon introducing each quantity (SFS, tMB and scMB) as short sentence explaining how it is measured in practice would help the non expert reader grasp the link to potential data pattern.

I was surprised by the model of mutation process upon division: (1.55) says that upon division, "each daughter cell acquires resistance with probability ν ", the same goes with 1.62 for neutral mutations: upon division each cell gets $\mu \sim \text{Poisson}(m)$ mutations. This is not classic to my knowledge (maybe it is equivalent) and I do not see how it is backed by biological knowledge. The most classic mutation model for asexual divisions is that, upon dna copy, the newly synthesized copy carries mutations ($\sim \text{poisson}(m)$) while the other (template) does not, one of the daughters inherits a mutation free template and the other carries potential mutations. Please clarify this important point (is it the same model in ref 16 ?).

I apologize for the delay I took to edit your ms, due to christmas holidays.

I hope you can address all the reviewers points (below), and my own remarks above, and thank you for your contribution to genetics.

Upon resubmission, please include:

1. A clean version of your manuscript;
2. A marked version of your manuscript in which you highlight significant revisions carried out in response to the major points raised by the editor/reviewers (track changes is acceptable if preferred);
3. A detailed response to the editor's/reviewers' feedback and to the concerns listed above. Please reference line numbers in this response to aid the editor and reviewers.

Your paper will likely be sent back out for review.

Additionally, please ensure that your resubmission is formatted for GENETICS
<https://academic.oup.com/genetics/pages/general-instructions>

Follow this link to submit the revised manuscript: Link Not Available

Sincerely,

Guillaume Martin

Associate Editor
GENETICS

Approved by:
Nicholas Barton
Senior Editor
GENETICS

Reviewer #1 :

In the study titled "On the patterns of genetic intra-tumor heterogeneity before and after treatment" by Stein and Werner, the authors investigate the dynamics of mutation accumulation in evolving tumor population before/after treatment setting in combination of shrinking, growing and no change in tumor sizes through birth-death processes.

The authors develop analytical tools to match simulation results and propose metrics with potential use in clinical setting.

The authors describe the use of SFS as a null hypothesis for homogeneity whereby the deviations from the expected SFS given the growth dynamics can be a proxy for heterogeneity in the tumor. However, the clinical applicability of the assumptions seem undermined, for instance, multiple subpopulations of cells with distinct heterogeneity could exist hence spatial sampling could be informative or large scale aneuploidies could result in different estimates of SFS in which case VAF might be a more informative metrics. The authors should expand their comments on the potential applicability and drawbacks of such tests under the simplifying assumptions.

Another related point would be the clone size distribution (CSD) which the author use to characterize the distribution of sizes of resistant clones emerging consecutively. However, the resistance cost does not seem to be taken into account. The authors should comment on whether under this assumption the clone size difference between the first and the second is overestimated? Also as a clinical association, how can this formalism relate to an adaptive-treatment setting where one of the aims would be to prevent emergence of multiple resistant clones hence the timing/frequency of treatment/drug-holidays could be adjusted to maximize the clone size differences between the first and second resistant clone.

Overall, I think the authors develop a rigorous framework to evaluate the mutational dynamics in evolving tumor populations. However, it is difficult to estimate how realistic some of the assumptions made hence how realistic the proposed hypothesis testing across the evaluated metrics. The authors should expand on the potential drawbacks in the discussion.

Some other concerns:

- Broadly, the work seems to be needing a stronger biological relevance -- there is a sense as I read it that it is a series of calculations without clear motivating purpose or unifying 'story'
- The conclusions are overstated at times: E.g. lines 300-303, claiming to have addressed the 'long debate on treatment resistance' -- but of course the model presented has a number of assumptions and is later sold as a 'null model' in the discussion (which clearly should not be used to settle a debate)
- I question whether the conclusions made after some of the analytical results are novel? (e.g. line 294-295: "Our calculations suggest that drugs suppressing the accumulation of new mutations - if feasible - would significantly reduce the risk of novel acquired genetic resistance in such patients, while at the same time weeding out preexisting genetic diversity")  did we need calculations to tell us this?
- I felt both the intro and discussion were lacking context and an explanation of biological relevance
- It would be nice for the authors to consider how sequencing technologies impact detectability of the distributions they describe (as these will have variable detection for the SNPs the authors are modeling, especially if they are more rare)

Reviewer #2 :

The authors study site frequency spectrum and mutation burden in a birth-death process with mutations describing cancer before and after treatment. The paper is missing background and references on related work on accumulation of mutations in birth-death processes. A few of these works are cited, but a clear description (in the introduction) of what has been done before and what is new in this work is needed.

Many of the results in the paper are relatively simple extensions of previous results (like deriving the expected site frequency spectrum in a population started by N_0 cells, following the previous proof which derived it for $N_0=1$). Similarly, their expression for expected total mutational burden is again derived in previous work.

Another example of not much novelty is the following:

"In an exponentially growing population, the first mutant clone has a large advantage as all later arriving clones need to catch up to become dominant. We thus asked whether the first resistant clone always dominates all other later arising resistant clones. Here, we show that this is not the case... " This is well known and noted in many previous works.

In sum, I am not convinced that this work is contributing much new insight.

Key references:

Durrett. Population genetics of neutral mutations in exponentially growing cancer cell populations. *Annals of Applied Probability* 2013.

Bozic, Gerold & Nowak. Quantifying clonal and subclonal mutations in cancer evolution. *PLoS Computational Biology* 2016.

Ohtsuki & Innan. Forward and backward evolutionary processes and allele frequency spectrum in a cancer cell population. *Theoretical Population Biology* 2017.

Cheek & Antal. Mutation frequencies in a birth-death branching process. *The Annals of Applied Probability* 2018.

Cheek & Antal. Genetic composition of an exponentially growing cell population. *Stochastic Processes and their Applications* 2020.

Gunnarsson, Leder & Zhang. Limit theorems for the site frequency spectrum of neutral mutations in an exponentially growing population. *arXiv* 2023.

Leder, Sun, Wang & Zhang. Parameter estimation from single patient, single time-point sequencing data of recurrent tumors. *Journal of Mathematical Biology* 2024.

Associate Editor Comments:

Response letter to the editor and reviewers

GENETICS-2024-307634

On the patterns of genetic intra-tumour heterogeneity before and after treatment

Associate Editor Comments:

Dear Dr. Stein:

Two experts in the field have reviewed your manuscript, and I have read it as well. The manuscript tackles a timely question, that is of interest beyond cancer biology, and the main figures show clear tests of the analytical predictions, which is appreciable. The structure with all proofs in appendix is a good strategy given the focus of genetics. Therefore, while your manuscript is not currently acceptable for publication in GENETICS, we would welcome a substantially revised manuscript. Both reviewers have comments and concerns to be addressed in a revised manuscript. You can read their reviews at the end of this email.

Dear Guillaume, we are grateful to you and both reviewers for the time to assess our manuscript and for the constructive suggestions! We apologize for the delay. Alex was finalizing his thesis in the last weeks. Please find a detailed response to all suggestions below.

Both reviewers point to important aspects that should all be tackled before resubmission. In particular, the links to existing theoretical results should be made clearer, and the biological limits of the model's simplifications in applying to actual sequence data in cancer. I would add to that second point that pharmacokinetics and pharmacodynamics may affect the outcomes compared to a simple fixed birth and death rate before/after treatment (which would find its place in the part dedicated to limits 1.491-500, a part that requires some extensions, see rev#1 & 2).

We agree and have now extended the introduction with a more elaborate description of previous theoretical analysis of genetic intra-tumour heterogeneity (lines 36-76). We also extended the discussion listing limitations of our results, including references to additional biological complexities such as pharmacokinetics and pharmacodynamics (lines 575-606). Throughout the text, we discuss possible connections to currently available sequencing strategies.

The introduction is fairly short and could be expanded (reasonably) with references to existing work on these questions, possibly to reviews in the case of large bodies of literature, rev#2 has provided several suggestions. As ref 16 provides the SFS for $N_0=1$, it should be stated from the onset that the results on SFS are extensions of this ref when $N_0 > 1$ (homogeneous), and when two clones with varying rates are present (resistant + sensitive). The same goes a priori for the tMB, where the link to existing result seems partially lacking (see rev#2). When dealing with standing variance in an expanding asexual (pre-treatment), reference to the fluctuation test analysis literature should be done (from the intro) as it has 80 years of theoretical development since the Lea Coulson treatment (of what I believe is the same process: I may be wrong, but then distinction should be made with this

important body of literature). This literature includes models of the problem when the mutations carry a cost before treatment (see rev#1's comment); this is sometimes called the Mandelbrot-Koch model (Mandelbrot 1974, Koch 1982) and has a long history.

These are very good suggestions that we happily take on board. Indeed, the Luria-Delbrück distribution is mathematically related to the SFS. The LD distribution is derived from a two-type model while the SFS is usually derived assuming infinite sites. By equating the mutation rate for resistance mutations to the mutation rate of passenger mutations, the LD distribution describes the expected SFS when relaxing the infinite-sites assumption (but typically taking the limit of very low mutation rates). This connection was made explicit by Cheek & Antal which we cite in the introduction now. Another way of seeing the connection between those quantities is to consider the different clones that make up the subpopulation of resistant cells. Then, the mathematical expression for the expected SFS approximates the clone size distribution underlying the total resistant subpopulation.

Lea & Coulson considered a pure-birth process with deterministic growth of the wildtype population. Here, we consider a fully stochastic birth-death process, which is more general and only has been better understood in the last decade, see for example in Kessler & Levine (2015) and Cheek & Antal (2020). We make sure to point out these connections early in the introduction (lines 45-58).

We agree with reviewer #2 and follow their advice to extend references to studies of SFS, tMB and scMB and add additional motivations.

As suggested, we cite ref. 16 on the onset of the sections on the SFS and the tMB. We added simulations including the cost of resistance to cover reviewer #1 comment and mention the cost of resistance again in the discussion.

Upon introducing each quantity (SFS, tMB and scMB) as short sentence explaining how it is measured in practice would help the non expert reader grasp the link to potential data pattern.

This is a good suggestion. We have added a sentence at the beginning of each corresponding section on currently available sequencing methods to measure these quantities. We also point out that the tMB currently is difficult to measure in actual tumour populations, but can be extrapolated from single-cell data.

I was surprised by the model of mutation process upon division: (l.55) says that upon division, "each daughter cell acquires resistance with probability ν ", the same goes with l.62 for neutral mutations: upon division each cell gets $\mu \sim \text{Poisson}(m)$ mutations. This is not classic to my knowledge (maybe it is equivalent) and I do not see how it is backed by biological knowledge. The most classic mutation model for asexual divisions is that, upon dna copy, the newly synthesized copy carries mutations ($\sim \text{poisson}(m)$) while the other (template) does not, one of the daughters inherits a mutation free template and the other carries potential mutations. Please clarify this important point (is it the same model in ref 16?).

This is a very good observation. Indeed, the model in ref. 16 and related works make similar assumptions. There is a biological and theoretical rationale.

Biologically, in the context of somatic evolution, it has been shown that mutation accumulation is a compound process with division-dependent and division-independent “background” mutations (Abascal et al. 2021). While indeed, division-induced mutations act on one strand, background mutations act on both, and mutations may fix in either daughter during division. Often, background mutations tend to dominate. These background mutations act clocklike and manifest as an apparent increase in the per-division mutation rate.

Theoretically, we indeed can show that if mutation rates are low, having a Poisson distribution for the mutation accumulation on each daughter cell is equivalent to having a Poisson distribution on just one daughter cell. If mutation rates are of magnitude 1 or higher per division differences may appear in some but not all measures. In expectation, as in Gunnarsson et al. (2022), the SFS remains unchanged as it only depends on the expected number of mutations acquired during a single division. The scMB distribution would change slightly (the variance of the distribution does depend on some of the underlying biology). In practice, a model as in ref. 16 agrees well with currently available bulk and single-cell sequencing data, but it would indeed be of great interest to analyze the fully stochastic compound process of division-dependent and independent mutation rates and we might take this up in future work. We discuss this now in Supplementary Section P, which we refer to in lines 108-111.

References:

- D Kessler and H Levine, Scaling solution in the large population limit of the general asymmetric stochastic Luria-Delbruck evolution process (2015) *Journal of statistical physics*
- D Cheek and T Antal, Genetic composition of an exponentially growing cell population (2020) *Stochastic processes and their Applications*
- Abascal et al. Somatic mutation landscape at single-molecule resolution (2021) *Nature*

Reviewer #1 :

In the study titled "On the patterns of genetic intra-tumor heterogeneity before and after treatment" by Stein and Werner, the authors investigate the dynamics of mutation accumulation in evolving tumor population before/after treatment setting in combination of shrinking, growing and no change in tumor sizes through birth-death processes.

The authors develop analytical tools to match simulation results and propose metrics with potential use in clinical setting.

We thank the reviewer for the time to assess our manuscript and the constructive feedback that allowed us to improve our manuscript.

The authors describe the use of SFS as a null hypothesis for homogeneity whereby the deviations from the expected SFS given the growth dynamics can be a proxy for heterogeneity in the tumor. However, the clinical applicability of the assumptions seem undermined, for instance, multiple subpopulations of cells with distinct heterogeneity could exist hence spatial sampling could be informative or large scale aneuploidies could result in different estimates of SFS in which case VAF might be a more informative metrics. The authors should expand their comments on the potential applicability and drawbacks of such tests under the simplifying assumptions.

The reviewer makes a good point. We agree that sampling biases are a limitation and that spatial multi-region sampling could at least partially resolve those. We now discuss this aspect and its impact on clinical applicability (lines 583-590).

The reviewer is correct that large-scale aneuploidies would change the patterns of the SFS. This can be circumvented, for example, by only considering diploid regions of genomes in question. We added a discussion after the homogeneity test, which we think makes our work more relatable to sequencing data (lines 266-278).

Another related point would be the clone size distribution (CSD) which the author use to characterize the distribution of sizes of resistant clones emerging consecutively. However, the resistance cost does not seem to be taken into account. The authors should comment on whether under this assumption the clone size difference between the first and the second is overestimated? Also as a clinical association, how can this formalism relate to an adaptive-treatment setting where one of the aims would be to prevent emergence of multiple resistant clones hence the timing/frequency of treatment/drug-holidays could be adjusted to maximize the clone size differences between the first and second resistant clone.

We agree that the cost of resistance is important for some treatment forms. In response, we ran additional simulations focusing on the effect of resistance costs on the clone size differences. Interestingly, our simulations suggest that the cost of resistance leads to clone sizes being more similar to each other than without the cost of resistance. We added a new supplementary table with a summary of the new simulations and commented on those results in the results section (see lines 488-490 and Table S5). Unfortunately, a fully analytical treatment of these questions goes beyond the scope of this already rather lengthy manuscript.

The reviewer makes an important point on possible applications of adaptive therapies, which we mention now in the Discussion (lines 603-605). A question we are very interested in as well. Adjusting treatment strategies to maximise clone size differences is an intriguing idea that certainly deserves further attention.

Overall, I think the authors develop a rigorous framework to evaluate the mutational dynamics in evolving tumor populations. However, it is difficult to estimate how realistic some of the assumptions made hence how realistic the proposed hypothesis testing across the evaluated metrics. The authors should expand on the potential drawbacks in the discussion.

This is a fair point, and we significantly expanded on these drawbacks in the discussion (lines 575-606). Our aim in this study was to extend the underlying population genetics theory of evolving tumour populations to certain treatment scenarios. Indeed, applying these theoretical results in a clinical setting has many challenges. In our experience though, connecting population genetics theory and sequencing data often works surprisingly well in somatic tissues.

Some other concerns:

- Broadly, the work seems to be needing a stronger biological relevance -- there is a sense as I read it that it is a series of calculations without clear motivating purpose or unifying 'story'

We understand the reviewer's point here. Our manuscript aims to present a theoretical framework and its application to draw a picture of genetic patterns of tumours before and after treatment. Once working on the effects of treatment, we thought it useful to put these results in the context of untreated populations (which are much better understood theoretically). As such, we hope that our manuscript is of interest to readers working on similar models and/or seeking an understanding of the effect of treatment on genetic intra-tumour heterogeneity.

Following the reviewer's and editor's suggestions, we expanded our introduction as well as discussions of how our results can be connected to sequencing data to make our manuscript more relatable to the underlying biology.

- The conclusions are overstated at times: E.g. lines 300-303, claiming to have addressed the 'long debate on treatment resistance' -- but of course the model presented has a number of assumptions and is later sold as a 'null model' in the discussion (which clearly should not be used to settle a debate)

Very fair point. We changed the statements accordingly and included additional references on this topic (lines 365-375).

- I question whether the conclusions made after some of the analytical results are novel? (e.g. line 294-295: "Our calculations suggest that drugs suppressing the accumulation of new mutations - if feasible - would significantly reduce the risk of novel acquired genetic resistance in such patients, while at the same time weeding out preexisting genetic diversity")  did we need calculations to tell us this?

This is an interesting point. In a sense, we agree with the reviewer that in retrospect this result is obvious. Given sufficient time, in a birth-death process of a constant population, the genealogy must reduce to a single most recent common ancestor that existed post-treatment. However, we were still surprised how fast pre-existing mutations became extinct, making it practically relevant and not only a theoretical concept. We adjusted our discussion accordingly (lines 361-364).

- I felt both the intro and discussion were lacking context and an explanation of biological relevance

We extended the introduction to better motivate our research and incorporate our work into existing literature (lines 36-76), and we extended the discussion to elaborate on biological/clinical limitations (lines 575-606).

- It would be nice for the authors to consider how sequencing technologies impact detectability of the distributions they describe (as these will have variable detection for the SNPs the authors are modeling, especially if they are more rare)

We now discuss how noise and sequencing coverage impact the detectability in the discussion (lines 575-583).

Reviewer #2 :

The authors study site frequency spectrum and mutation burden in a birth-death process with mutations describing cancer before and after treatment. The paper is missing background and references on related work on accumulation of mutations in birth-death processes. A few of these works are cited, but a clear description (in the introduction) of what has been done before and what is new in this work is needed.

We thank the reviewer for the time to assess our manuscript and the constructive feedback that allowed us to improve our manuscript. We have now made significant additions to our introduction discussing the prior theoretical literature (lines 36-85).

Many of the results in the paper are relatively simple extensions of previous results (like deriving the expected site frequency spectrum in a population started by N_0 cells, following the previous proof which derived it for $N_0=1$). Similarly, their expression for expected total mutational burden is again derived in previous work.

We agree with the reviewer, eqn. (6) and eqn. (15) are (relatively simple) extensions of the results derived by Gunnarsson et al. (2021). To make this more clear, we cite Gunnarsson et al. at the onset of the SFS and tMB section.

Notably, besides the mentioned extension, eqn. (7) and eqn. (16) are novel to the best of our knowledge and play a key role in determining the site frequency spectrum and total mutational burden after treatment.

Another example of not much novelty is the following:

"In an exponentially growing population, the first mutant clone has a large advantage as all later arriving clones need to catch up to become dominant. We thus asked whether the first resistant clone always dominates all other later arising resistant clones. Here, we show that this is not the case..." This is well known and noted in many previous works.

We thank the reviewer for pointing this out. After reading through the suggested paper list below, we found relevant results in Bozic et al. (2018) and the related article Bozic & Nowak (2014) who derived the size distribution of clones ordered by their appearance, and Cheek & Antal (2018) who also derived an expression for size distribution of the largest clone. We cite and discuss those results now (lines 496-498 and 511-515 in the main text and additions in Supplementary Section N).

In sum, I am not convinced that this work is contributing much new insight.

We thought it to be useful to embed results of treatment effect on intra-tumour heterogeneity within better-known results of untreated growing populations. As such, our manuscript contains novel and known results. We wish to point out that many of the presented results are novel (e.g. eqn. 12 or eqn. 20), present alternative derivations for known results (e.g. eqn. 9), put known analytic results into a new context (e.g. eqn. 11) or refine known theoretical results (e.g. eqn. 24). We hope that these distinctions become more clear in the revised manuscript.

Key references:

Durrett. Population genetics of neutral mutations in exponentially growing cancer cell populations. *Annals of Applied Probability* 2013.

Bozic, Gerold & Nowak. Quantifying clonal and subclonal mutations in cancer evolution. *PLoS Computational Biology* 2016.

Ohtsuki & Innan. Forward and backward evolutionary processes and allele frequency spectrum in a cancer cell population. *Theoretical Population Biology* 2017.

Cheek & Antal. Mutation frequencies in a birth-death branching process. *The Annals of Applied Probability* 2018.

Cheek & Antal. Genetic composition of an exponentially growing cell population. *Stochastic Processes and their Applications* 2020.

Gunnarsson, Leder & Zhang. Limit theorems for the site frequency spectrum of neutral mutations in an exponentially growing population. *arXiv* 2023.

Leder, Sun, Wang & Zhang. Parameter estimation from single patient, single time-point sequencing data of recurrent tumors. *Journal of Mathematical Biology* 2024.

We included missing references in the revised manuscript.

We thank the reviewer for the helpful feedback and important references that led to a better incorporation into the existing theoretical literature.

References:

- I Bozic and M Nowak, Timing and heterogeneity of mutations associated with drug resistance in metastatic cancers (2014) *PNAS*

May 13, 2025

RE: GENETICS-2025-308016

Mr. Alexander Stein
Barts Cancer Institute
Centre for Cancer Genomics and Computational Biology
John Vane Science Centre
Charterhouse Square
London EC1M 6AU
United Kingdom

Dear Dr. Stein:

Congratulations, your manuscript titled "On the patterns of genetic intra-tumour heterogeneity before and after treatment" is accepted for publication in GENETICS! Many thanks for submitting your research to the journal.

Both reviewers and I are very happy with the modifications and the resulting article will be a nice treatment of resistance evolution in general, making interesting connection between important points in the literature and hopefully also different applications (cancer, bacteria, viruses, fungi). Congratulation to the first author on finishing their thesis.

To Proceed to Publication:

1. Format your article according to GENETICS style: <https://academic.oup.com/genetics/pages/general-instructions>
2. Ensure that you comply with data and community resource citation guidelines: <https://academic.oup.com/genetics/pages/general-instructions#Data-Policy>
3. Upload your final files at <https://genetics.msubmit.net>
4. Add oupsupport@scipris.com and genetics.oup@novatechset.com (or the domains @scipris.com and @novatechset.com) to your email program's "safe senders" list. You will be contacted by both at various points during the production process.

Notes:

- Your currently-accepted manuscript (unedited, as submitted, reviewed, and accepted) will be published at GENETICS and deposited into PubMed as an Advance Access article. Notify sourcefiles@thegsajournals.org before signing your license if you do not wish to publish your article via Advance Access.
- We invite you to submit an original color figure related to your paper for consideration as cover art. Please email your submission to the editorial office or upload it with your final files. You can submit a small-sized image for evaluation, and if selected, the final image must be a TIFF file 2513px wide by 3263px high (8.375 by 10.875 inches; resolution of 600ppi). Please avoid graphs and small type.
- After files are sent to Oxford University Press we use SciPris to manage article licensing and payment. If you do not have a SciPris account, you will receive an email from no-reply@scipris.com to sign up to use Oxford University Press' author portal. After logging in, follow the online instructions to sign your license and arrange any payment due.

If you have any questions or encounter any problems while uploading your accepted manuscript files, please email the editorial office at sourcefiles@thegsajournals.org.

Sincerely,

Guillaume Martin
Associate Editor
GENETICS

Approved by:
Nicholas Barton
Senior Editor
GENETICS

Review comments (if applicable):

Reviewer #1 :

Kudos to the authors for a great paper, and a thorough revision. I also imagine that congratulations are involved to Mr. (now Dr?) Stein on his defense, as this paper certainly warrants a degree.

Reviewer #2 :

The manuscript is now significantly stronger as it is aware of existing literature and it is clearer what findings are novel.